# New bounded unit Weibull model: Applications with quantile regression

**Laxmi Prasad Sapkota**[1,2]*, **Nirajan Bam**[2], **Vijay Kumar**[2,3]

**1** Department of Statistics, Tribhuvan University, Tribhuvan Multiple Campus, Tansen, Nepal,
**2** Department of Mathematical and Physical Sciences, Miami University, Hamilton, Ohio, United States of America, **3** Department of Mathematics and Statistics, DDU Gorakhpur University, Gorakhpur, Uttar Pradesh, India

* laxmisapkota75@gmail.com

**Data availability statement:** All relevant data are within the paper and its Supporting information files. Additionally, these datasets are freely accessible via the following links: Educational Attainment Data (https://data-explorer.oecd.org) and Risk Assessment Data

## Abstract

In practical scenarios, data measurements like ratios and proportions often fall within the 0 to 1 range, posing unique modeling challenges. While beta and Kumaraswamy distributions are widely used, alternative models often yield better performance, though no clear consensus exists. This paper introduces a new bounded probability distribution based on a transformation of the Weibull distribution, with properties such as moments, entropies, and a quantile function. Additionally, we have developed the sequential probability ratio test (SPRT) for the proposed model. The maximum likelihood estimation method was employed to estimate the model parameters. A Monte Carlo simulation was conducted to evaluate the performance of parameter estimation for the model. Finally, we formulated a quantile regression model and applied it to data sets related to risk assessment and educational attainment, demonstrating its superior performance over alternative regression models. These results highlight the importance of our contributions to enhancing the statistical toolkit for analyzing bounded variables across different scientific fields.

## Introduction

In many real-world applications, data are restricted to a bounded interval, often within the (0, 1) range, representing proportions, ratios, or fractions. The bounded nature of such data presents unique modeling challenges, leading researchers to focus on addressing these issues. The beta distribution and the Kumaraswamy distribution are frequently used in scenarios where outcomes are measured within the unit interval. Notably, the absence of a closed-form expression for the cumulative distribution function (CDF) of the beta distribution complicates the analysis of statistical properties that depend on the quantile function. Additionally, the beta distribution is not well-suited for sparse data scenarios [1]. To address these limitations, Kumaraswamy introduced the Kumaraswamy distribution as an alternative to the beta distribution. However, the Kumaraswamy distribution also lacks closed-form expressions for moments such as the mean and variance, posing challenges in the analytical study of statistical properties related to moments[2,3]. Thus, there is growing attention in research on introducing new distributions as alternatives to the beta and Kumaraswamy distributions.

(https://instruction.bus.wisc.edu/jfrees/jfreesbooks/Regression%20Modeling/BookWebDec2010/data.html).

**Funding:** The author(s) received no specific funding for this work.

**Competing interests:** The authors have declared that no competing interests exist.

In recent years, several distributions have been introduced as an alternative to the beta distribution and its associated regression models. Examples include the unit-Birnbaum-Saunders distribution [4], the two-parameter unit Gaussian distribution [5], the unit Johnson $S_U$ distribution [6], the two-parameter log-cosine-power unit distribution [7], and the unit-Lindley distribution [8], unit-gamma distribution [9], and the unit-exponentiated Lomax distribution [10]. Some of these distributions have more than two parameters, such as the unit-exponentiated Lomax distribution, which increases model complexity, while others, like the one-parameter unit Lindley distribution, offer less flexibility in controlling the precision of the distribution. More importantly, all of these distributions overlook a crucial statistical property known as the SPRT.

In applied research, researchers often require regression models where the response variable is bounded within the interval (0, 1). Early work on bounded regression, particularly in the context of police behavior, was conducted by Johns and Gates (1993) [11]. Subsequently, Paolino (2001) introduced a simplified version of beta regression, providing a more practical approach for such applications [12]. However, applying a regression framework to bounded data gained significant attention following the introduction of beta regression by Ferrari and Cribari-Neto (2004) [13]. The key distinction between Paolino's and Ferrari's work is that Paolino models both the conditional mean and the precision parameter as outcomes, while Ferrari and Cribari-Neto treat the precision parameter as a nuisance. Although beta regression has gained popularity for its flexibility in modeling continuous data bounded between 0 and 1, it is highly sensitive to outliers. Some conditional mean-based bounded regression models that have been proposed as alternatives to beta regression include unit Lindley regression [8], log-weighted exponential regression [14], and log-Bilal regression [15]. When the response variable exhibits skewness or contains outliers, traditional mean-based regression models may not be adequate to infer the relationship between the outcome variable and predictors, as the mean is highly sensitive to outliers. Consequently, median-based regression is suitable in cases where the response variable shows skewness or includes outliers, as the median is robust against outliers (see [16,17]).

Quantile regression [17] was first presented by Koenker and Bassett as a generalized linear framework to model the conditional quantiles of a response variable [18]. Its robustness to outliers has led to a surge in its popularity within both applied and theoretical statistics over the past few decades. Recently, there has been a growing interest in using quantile regression to model unit-bounded response variables along with sets of covariates. The extension of quantile regression to model bounded response variables and covariates is driven by two main reasons. Firstly, some distributions do not have a closed form for their mean but do have closed forms for their quantiles. Thus, using quantile regression with the quantile function simplifies the incorporation of regression structures in the proposed model. Another important factor is that replacing the conditional mean with the median is one of the most effective ways to reduce the influence of outliers, which is achievable through quantile regression.

Given the difficulties in fitting the beta distribution in engineering and hydrology applications, Kumaraswamy introduced the Kumaraswamy distribution as an alternative [1]. Nevertheless, its use was initially restricted due to the lack of closed-form moments. In 2013, Mitnik and Baek enhanced the Kumaraswamy distribution by reparametrizing it for quantile regression, which increased interest in Kumaraswamy regression [3]. Since then, quantile regression in bounded-area distributions has become more popular. For example, recent advancements in unit interval quantile regressions include the unit-Weibull distribution and its quantile regression [19], the two-parameter Burr XII distribution and its quantile regression [20], the log-log unit distribution and its quantile regression [21], the half-normal unit distribution and its quantile regression [22], the log-logistic unit distribution and its quantile

regression [23], unit Burr XII quantile regression model [24], and the unit chain distribution and its quantile regression [25].

Although various regression models are discussed in the literature, some are designed to model the conditional mean, and others are designed to model the conditional median of bounded response variables. There is no agreement on a viable alternative to the beta and Kumaraswamy regression. While quantile-based regression is generally more robust to outliers than conditional mean-based regression, the widely used Kumaraswamy quantile regression remains susceptible to extreme values, particularly when the distribution exhibits unimodal behavior [26]. To address this limitation, [26] introduced the three-parameter Kumaraswamy Rectangular distribution, which enhances robustness. However, the inclusion of an additional parameter increases model complexity, making estimation and interpretation more challenging. Several other three-parameter probability distributions defined on the unit interval, along with their associated quantile regression models—such as the unit-exponentiated Lomax distribution [10] and the bounded exponentiated Weibull distribution [27]—further increase the complexity of statistical modeling.

Several unit-interval bounded probability distributions, along with their corresponding regression and quantile regression models have been discussed above, each with its own strengths and limitations. However, a fundamental property that all such distributions lack is the SPRT. SPRT is a key property of probability distributions with immediate applications in various applied fields. Some of its potential advantages include: it allows researchers to stop data collection early once sufficient evidence has been gathered to make a decision, thereby saving both time and resources [28]. Additionally, SPRT requires significantly smaller sample sizes compared to traditional hypothesis tests, making it more efficient in terms of both cost and data collection efforts [29]. Despite the importance of the SPRT in applied fields, it has not yet been introduced in the context of unit-interval bounded probability distributions. This gap underscores the need for a new, flexible probability distribution that not only supports an associated quantile regression model resilient to extreme values in bounded response variables but also integrates key statistical properties like SPRT, thereby enhancing its practical applicability and efficiency. Our proposed model and its quantile regression exhibit several benefits and improvements over existing models. Thus, the primary objective of this study is to introduce new bounded probability distributions and associated quantile regression models and to present essential properties, including the SPRT.

The main advantage of the new bounded Weibull distribution lies in its flexibility, unimodal nature, and the bathtub-shaped probability density function (PDF) and hazard rate function, featuring several key properties not found in other unit interval bounded distributions. For instance, the new bounded Weibull distribution has only two parameters. It provides closed-form expressions for the CDF and quantile functions, with its PDF being expressible as linear densities of the beta distribution. This feature enables us to introduce a new bounded quantile regression model as a viable alternative to the widely used beta [13] and Kumaraswamy regression [3] models. Furthermore, the inclusion of the SPRT, absent in other bounded data models, is a distinctive feature of our proposed approach. SPRT enables real-time hypothesis testing, allowing decisions to be made during data collection instead of waiting for the complete datasets, thus enhancing efficiency and practicality [28].

After introducing the PDF, CDF, quantile function, and several essential properties of the new bounded unit-Weibull distribution, we proceeded to validate it and its associated quantile regression through simulation studies. Furthermore, we assessed its applicability by applying it to two real datasets. One data set originates from a managerial cost-effectiveness study conducted by [30]. This study measures the variable firm cost that evaluates cost effectiveness,

representing the total property and casualty premiums along with uninsured losses as a percentage. The variable is bounded within the range of 0 and 1 and exhibits outliers (see Fig 1). Another data set pertains to student achievement. Data on student achievement and associated predictor variables for OECD countries and other nations are available on the OECD website. Numerous studies have analyzed educational attainment in OECD countries using various regression models [31–33]. Outliers may manifest in both datasets. Traditional regression approaches may violate critical assumptions, and attempts to rectify these assumptions may infringe upon the bounded nature of the response variable. Although beta regression may offer advantages over classical regression methods, it typically models the mean as the response variable, rendering it susceptible to outliers and potentially less robust in such scenarios. In addition, Kumaraswamy quantile regression is also not robust to extreme values, particularly when the data distribution exhibits unimodal behavior [26]. Therefore, we used our proposed quantile regression model in both aforementioned data scenarios and compared its performance to beta regression, Kumaraswamy quantile regression, and several other similar quantile regressions.

The paper is organized as follows. In Section New Bounded Unit Weibull (NBUW) model, we formulate a new unit distribution. Section Some properties of NBUW model examines some properties of the new model. In Section NBUW quantile regression model, we present the new parameterized model and its quantile regression model. We conducted an extensive Monte Carlo simulation study to evaluate the performance of the emphasized estimators in Section Simulation. Then, we present the application of the regression model using two real-world data examples to illustrate the usefulness of our model in modeling real-world data in Section Application. Finally, conclusions are given in Section Conclusion.

## New Bounded Unit Weibull (NBUW) model

The unit Weibull distribution was initially introduced by modifying the PDF and CDF of the Weibull distribution through the transformation $Y = e^{-X}$ [34]. In contrast, this research uses a different transformation technique as outlined in [19], where the transformation $X = \frac{1}{1+Y}$

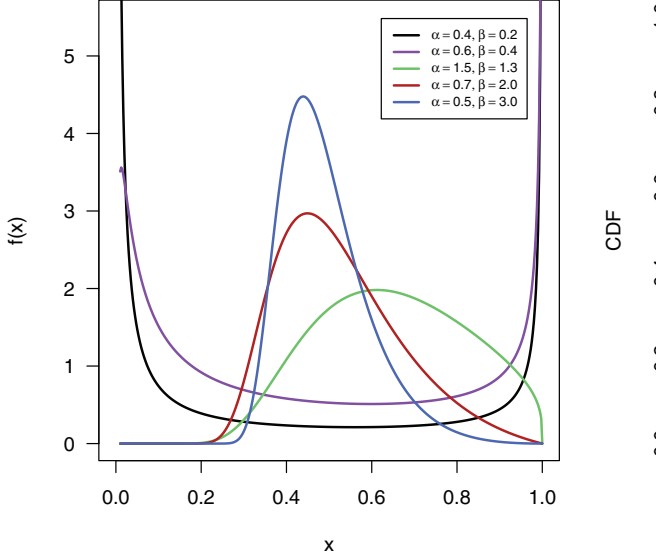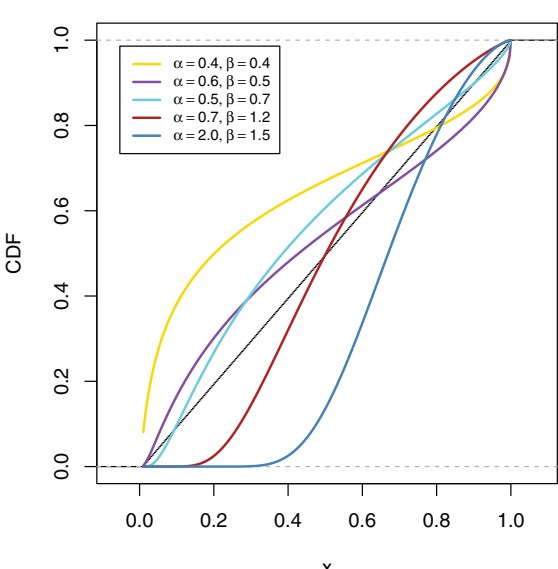

**Fig 1. Different curves of PDF and CDF of NBUW distribution.**

is applied to transform the Weibull $(0, \infty)$ to Weibull $(0,1]$ distribution. Here, $Y$ follows a Weibull distribution with its corresponding PDF and CDF.

$$F(y; \alpha, \beta) = 1 - \exp(-\alpha x^{\beta}); \ y > 0, \alpha, \beta > 0. \tag{1}$$

$$f(y; \alpha, \beta) = \alpha \beta x^{\beta-1} \exp(-\alpha x^{\beta}); \ y > 0, \alpha, \beta > 0. \tag{2}$$

By transforming Eqs (1) and (2), we have defined the new bounded Weibull model whose CDF and PDF are presented as follows.

$$F(x; \alpha, \beta) = e^{-\alpha \left( \frac{1-x}{x} \right)^{\beta}}; 0 < x \leq 1, \alpha, \beta > 0. \tag{3}$$

$$f(x; \alpha, \beta) = \alpha \beta \left( \frac{1-x}{x} \right)^{\beta-1} x^{-2} e^{-\alpha \left( \frac{1-x}{x} \right)^{\beta}}; 0 < x \leq 1, \alpha, \beta > 0. \tag{4}$$

Various shapes of the CDF and PDF of the NBUW distribution are displayed in Fig 1. It is observed that the PDF can exhibit bathtub, right-skewed, and unimodal shapes.

**Quantile function:** The quantile function $Q_q$ of the NBUW model can be presented as

$$Q_q(\alpha, \beta) = \left[ 1 + \left( \frac{-log(q)}{\alpha} \right)^{1/\beta} \right]^{-1}; q \in (0, 1). \tag{5}$$

**Hazard rate function (HRF):** To model the quality or survival rate of an item or object in fields such as engineering or environmental studies, the HRF can be utilized. The HRF for the NBUW distribution is given by:

$$h(x; \alpha, \beta) = \alpha \beta \left( \frac{1-x}{x} \right)^{\beta-1} x^{-2}. \tag{6}$$

## Some properties of NBUW model

### 0.1 Linear form of NBUW distribution

Using the Binomial and exponential expansions of the PDF defined in Eq (4), we can express it into a linear form of densities of beta distribution (first kind) as

$$
\begin{aligned}
f(x; \alpha, \beta) &= \alpha \beta \sum_{i=0}^{\infty} \sum_{j=0}^{\infty} \Delta_{ij} \ x^{i-\beta j - \beta - 1}(1-x)^{\beta j} \\
&= \alpha \beta \sum_{i=0}^{\infty} \sum_{j=0}^{\infty} \Delta_{ij} \ h(x; i - \beta j - \beta, \ \beta j + 1).
\end{aligned} \tag{7}
$$

where $h(x)$ is the PDF of the beta distribution (first kind) with parameters $(i - \beta j - \beta)$ and $(\beta j + 1)$ and $\Delta_{ij} = \frac{(-1)^{i+j}}{j!} \binom{\beta-1}{i} \alpha^j$.

## 0.2 Moments of NBUW distribution

The $m^{th}$ moment of $X \sim NBUW(\alpha, \beta)$ about the origin can be calculated as

$$\mu'_m = \alpha\beta \sum_{i=0}^{\infty} \sum_{j=0}^{\infty} \Delta_{ij} \ B(i + r - \beta j - \beta, \ \beta j + 1). \tag{8}$$

where $B(a, b) = \int_0^1 x^{a-1}(1 - x)^{b-1} dx$ is the beta function. Using Eq (8), we can compute the mean and variance of the NBUW distribution as follows.

$$\mu'_1 = \alpha\beta \sum_{i=0}^{\infty} \sum_{j=0}^{\infty} \Delta_{ij} \ B(i - \beta j - \beta + 1, \ \beta j + 1).$$

and the second moment is

$$\mu'_2 = \alpha\beta \sum_{i=0}^{\infty} \sum_{j=0}^{\infty} \Delta_{ij} \ B(i - \beta j - \beta + 2, \ \beta j + 1).$$

Hence, the variance of X of the NBUW can be calculated as

$$V(X) = \mu'_2 - (\mu'_1)^2$$
$$= \alpha\beta \sum_{i=0}^{\infty} \sum_{j=0}^{\infty} \Delta_{ij} \ B(i - \beta j - \beta + 2, \ \beta j + 1) -$$
$$\left\{ \alpha\beta \sum_{i=0}^{\infty} \sum_{j=0}^{\infty} \Delta_{ij} \ B(i - \beta j - \beta + 2, \ \beta j + 1) \right\}^2.$$

Using the above expressions for mean and variance, we have presented the graphical view of the mean and variance for different values of the shape parameter $\beta$ in Fig 2.

### Moment Generating Function (MGF)

Using the expression for moments Eq (8), we can derive the MGF of the NBUW model as

$$M_X(\eta) = \alpha\beta \sum_{i=0}^{\infty} \sum_{j=0}^{\infty} \sum_{k=0}^{\infty} \frac{\eta^k}{k!} \Delta_{ij} \ B(i + r - \beta j - \beta, \ \beta j + 1).$$

### Entropy

Entropy is an important statistical tool to obtain the randomness associated with an event or a system. There are so many types of entropies, here for the NBUW model, we have presented two types of entropies as follows.

**Renyi entropy**

The Renyi entropy can be defined as

$$RE_\theta(x) = \frac{1}{1 - \theta} \log \left[ \int_0^1 \{f(x)\}^\theta dx \right]; \ \theta > 0, \theta \neq 1. \tag{9}$$

For this we first need to find $\{f(x)\}^\theta$, where $f(x)$ is the PDF defined in Eq (4) and can be obtained as

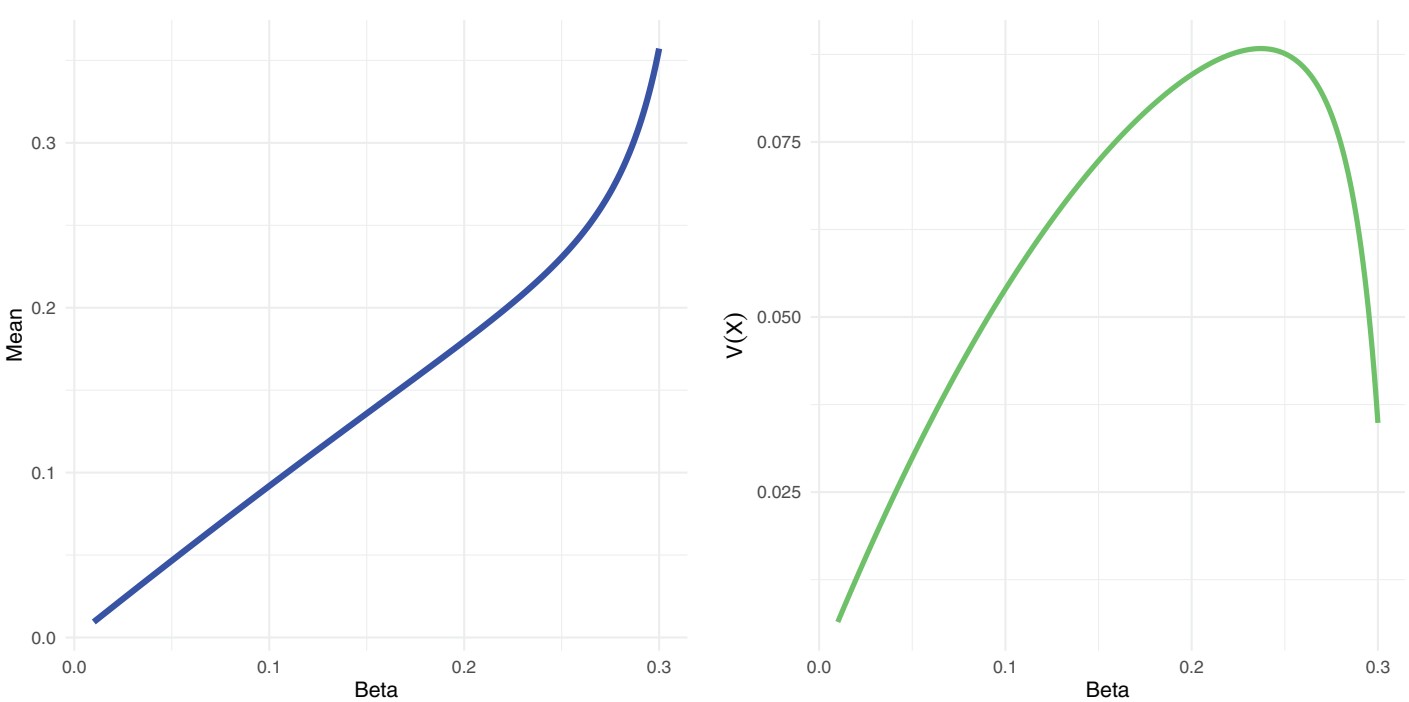

**Fig 2. Mean and variance plots of NBUW model for different values of the shape parameter $\beta$.**

$$\{f(x)\}^\theta = \sum_{i=0}^{\infty}\sum_{j=0}^{\infty}\Delta_{ij}^*(\alpha\theta)^j\binom{\beta\theta-\theta}{i}x^{i-\beta j-\beta\theta-\theta}(1-x)^{\beta j}. \tag{10}$$

where $\Delta_{ij}^* = \frac{(-1)^{i+j}}{j!}\binom{\beta\theta-\theta}{i}(\alpha\theta)^j(\alpha\beta)^\theta$. Now substituting the Eq (10) in Eq (9), we get

$$
\begin{aligned}
RE_\theta(x) &= \frac{1}{1-\theta}\log\left[\sum_{i=0}^{\infty}\sum_{j=0}^{\infty}\Delta_{ij}^*\int_0^1 x^{i-\beta j-\beta\theta-\theta}(1-x)^{\beta j}dx\right]\\
&= \frac{1}{1-\theta}\log\left[\sum_{i=0}^{\infty}\sum_{j=0}^{\infty}\Delta_{ij}^*B(i-\beta j-\beta\theta-\theta,\beta j+1)\right].
\end{aligned}\tag{11}
$$

**q-entropy**

Similarly, q-entropy for the NBUW distribution can be calculated using the Renyi entropy 11 as

$$
\begin{aligned}
qE_q(x) &= \frac{1}{1-q}\log\left[1-\int_0^1\{f(x)\}^q dx\right];\ q>0, q\neq 1\\
&= \frac{1}{1-\theta}\log\left[1-\sum_{i=0}^{\infty}\sum_{j=0}^{\infty}\Delta_{ij}^*B(i-\beta j-\beta\theta-\theta,\beta j+1)\right].
\end{aligned}
$$

## Sequential Probability Ratio Test (SPRT)

Let X have a $NBUW(\alpha,\beta)$ where unknown shape parameter $\beta\in\{\beta_0,\beta_1\}$ where $\beta_1>\beta_0$ and $\alpha$ is known scale parameter (say). Now we have the SPRT for testing the following

hypothesis. **Null hypothesis** $(H_0) : \beta = \beta_0$ against **Alternative hypothesis** $(H_1) : \beta = \beta_1$. Then the sequential likelihood ratio statistic is calculated as $\log(\lambda_n) = \sum_{i=1}^{n} \xi_i$, where

$$\xi_i = \log\left[\frac{f(x_i, \beta_1)}{f(x_i, \beta_0)}\right]$$

$$= \log\left(\frac{\beta_1}{\beta_0}\right) + (\beta_1 - \beta_0)\log\left(\frac{1-x_i}{x_i}\right) - \alpha\left(\frac{1-x_i}{x_i}\right)^{\beta_1} + \alpha\left(\frac{1-x_i}{x_i}\right)^{\beta_0}.$$

Hence, the SPRT statistic can be computed as

$$\log(\lambda_n) = n\log\left(\frac{\beta_1}{\beta_0}\right) + (\beta_1 - \beta_0)\sum_{i=1}^{n}\log\left(\frac{1-x_i}{x_i}\right) - \alpha\sum_{i=1}^{n}\left(\frac{1-x_i}{x_i}\right)^{\beta_1} + \alpha$$
$$\sum_{i=1}^{n}\left(\frac{1-x_i}{x_i}\right)^{\beta_0}.$$

For the SPRT test, we can decide the acceptance or rejection of $H_0$ as follows.

1. Accept $H_0$ if $\log(\lambda_n) \leq \log B$

$$n\log\left(\frac{\beta_1}{\beta_0}\right) + (\beta_1 - \beta_0)\sum_{i=1}^{n}\log\left(\frac{1-x_i}{x_i}\right) - \alpha\sum_{i=1}^{n}\left(\frac{1-x_i}{x_i}\right)^{\beta_1} +$$
$$\alpha\sum_{i=1}^{n}\left(\frac{1-x_i}{x_i}\right)^{\beta_0} \leq \log\left(\frac{\psi}{1-\delta}\right).$$

2. Reject $H_0$ if $\log(\lambda_n) \geq \log A$

$$n\log\left(\frac{\beta_1}{\beta_0}\right) + (\beta_1 - \beta_0)\sum_{i=1}^{n}\log\left(\frac{1-x_i}{x_i}\right) - \alpha\sum_{i=1}^{n}\left(\frac{1-x_i}{x_i}\right)^{\beta_1} +$$
$$\alpha\sum_{i=1}^{n}\left(\frac{1-x_i}{x_i}\right)^{\beta_0} \geq \log\left(\frac{1-\psi}{\delta}\right).$$

3. Continue sampling to examine an additional observation if $\log B < \log(\lambda_n) < \log A$

$$\log\left(\frac{\psi}{1-\delta}\right) < n\log\left(\frac{\beta_1}{\beta_0}\right) + (\beta_1 - \beta_0)\sum_{i=1}^{n}\log\left(\frac{1-x_i}{x_i}\right) - \alpha\sum_{i=1}^{n}\left(\frac{1-x_i}{x_i}\right)^{\beta_1} +$$
$$\alpha\sum_{i=1}^{n}\left(\frac{1-x_i}{x_i}\right)^{\beta_0} < \log\left(\frac{1-\psi}{\delta}\right).$$

Here, A and B are boundary points and can be calculated as

$$\delta = Prob(\text{type-I error}), \quad \psi = Prob(\text{type-II error}),$$
$$A \approx \frac{1-\psi}{\delta}, \quad B \approx \frac{\psi}{1-\delta}, \quad \text{and } 0 < B < 1 < A.$$

**Average Sample Number (ASN) function** If the same SPRT is carried out repeatedly many times, then we shall get different values of the random variable N, the number of observations needed to decide on the SPRT. The average value of N to reach a decision is called ASN, and it is denoted by $E(N)$. Since ASN depends upon the parameter to be tested, it is also called the ASN function and is denoted by $E_\beta(N)$ for this study. Consider an SPRT of strength $(\delta, \psi)$ and boundary points $(A,B)$ for testing $(H_0) : \beta = \beta_0$ against $(H_1) : \beta = \beta_1$. Let $x_1, x_2, \ldots$ be a sequence of i.i.d. observations sampled from $NBUW(\alpha, \beta)$ for the SPRT, then we can obtain the ASN as

$$E(N) = \frac{L(\beta) \log B + (1 - L(\beta)) \log A}{E(\xi)},$$

where $L(\beta)$ is the operating characteristic (OC) function of SPRT and it can be obtained under $H_0$ as

$$L(\beta_0) = Prob(\text{Accept } H_0/H_0 \text{ is true})$$
$$= 1 - \delta.$$

The ASN required to terminate the SPRT under $H_0$ is given by

$$E_{\beta_0}(N) = \frac{L(\beta_0) \log B + (1 - L(\beta_0)) \log A}{E_{\beta_0}(\xi)}$$
$$= \frac{\delta \log\left(\frac{1-\psi}{\delta}\right) + (1 - \delta) \log\left(\frac{\psi}{1-\delta}\right)}{E_{\beta_0}(\xi)}.$$

provided that $E_{\beta_0}(\xi) \neq 0$. Similarly, under $H_1$ is given by

$$E_{\beta_1}(N) = \frac{L(\beta_1) \log B + (1 - L(\beta_1)) \log A}{E_{\beta_1}(\xi)}$$
$$= \frac{\delta \log\left(\frac{\psi}{1-\delta}\right) + (1 - \psi) \log\left(\frac{1-\psi}{\delta}\right)}{E_{\beta_1}(\xi)}.$$

provided that $E_{\beta_1}(\xi) \neq 0$.

## Simulation study

In this section, we employed Monte Carlo simulations to examine the finite-sample properties of Maximum Likelihood Estimates (MLEs) and asymptotic confidence intervals for parameters in the NBUW distribution. We evaluated the simulations by estimating bias (B), mean squared error (M), and confidence interval (CI) for the estimated parameters. We chose sample sizes of $n = 20, 45, 70, 95, 120, 145, 170,$ and $195$, encompassing various combinations of the parameters $\alpha$ and $\beta$. A fixed number of Monte Carlo replications were performed, with $N = 1000$. All simulations were carried out using R software, utilizing the maxLik package [35] with the BFGS algorithm to compute the MLEs of $\alpha$ and $\beta$. The summarized results of these simulations are presented in Tables 1, 2, 3, 4, 5, and 6.

**Table 1. Bias, MSE, and CI for MLE estimates** $\alpha = 0.5$ **and** $\beta = 0.75$.

| n | $B_\alpha$ | $B_\beta$ | $M_\alpha$ | $M_\beta$ | $CI_\alpha$ | $CI_\beta$ |
|---|---|---|---|---|---|---|
| 20 | 0.0012 | 0.0634 | 0.0249 | 0.0290 | (0.2309 0.8470) | (0.5820 1.1860) |
| 45 | -0.0105 | 0.0300 | 0.0102 | 0.0097 | (0.3092 0.7002) | (0.6260 0.9855) |
| 70 | -8.0E-04 | 0.0160 | 0.0068 | 0.0054 | (0.3583 0.6676) | (0.6381 0.9211) |
| 95 | 0.0014 | 0.0122 | 0.0045 | 0.0039 | (0.3771 0.6361) | (0.6492 0.8899) |
| 120 | 0.0024 | 0.0069 | 0.0038 | 0.0029 | (0.3857 0.6222) | (0.6585 0.8621) |
| 145 | 0.0019 | 0.0092 | 0.0029 | 0.0026 | (0.3958 0.6050) | (0.6690 0.8664) |
| 170 | -0.0033 | 0.0082 | 0.0027 | 0.0022 | (0.4024 0.5964) | (0.6747 0.8552) |
| 195 | -5.0E-04 | 0.0077 | 0.0021 | 0.0018 | (0.4150 0.5911) | (0.6804, 0.8425) |

**Table 2. Bias, MSE, and CI for MLE estimates** $\alpha = 0.75$ **and** $\beta = 0.5$.

| n | $B_\alpha$ | $B_\beta$ | $M_\alpha$ | $M_\beta$ | $CI_\alpha$ | $CI_\beta$ |
|---|---|---|---|---|---|---|
| 20 | 0.0098 | 0.0438 | 0.0444 | 0.0135 | (0.4051 1.2115) | (0.3764 0.7928) |
| 45 | 0.0038 | 0.0165 | 0.0182 | 0.0043 | (0.5102 1.0582) | (0.4172 0.6623) |
| 70 | 3.0E-04 | 0.0091 | 0.0102 | 0.0024 | (0.5725 0.9648) | (0.4217 0.6098) |
| 95 | 3.0E-04 | 0.0066 | 0.0082 | 0.0018 | (0.5871 0.9368) | (0.4318 0.5972) |
| 120 | 0.0013 | 0.0058 | 0.0064 | 0.0014 | (0.6017 0.9083) | (0.4369 0.5829) |
| 145 | 0.0058 | 0.0026 | 0.0047 | 0.0010 | (0.6275 0.8956) | (0.4472 0.5671) |
| 170 | 1.0E-04 | 0.0039 | 0.0044 | 0.0010 | (0.6277 0.8870) | (0.4500 0.5741) |
| 195 | 0.0031 | 0.0031 | 0.0036 | 9.0E-04 | (0.6351 0.8689) | (0.4498 0.5642) |

**Table 3. Bias, MSE, and CI for MLE estimates** $\alpha = 1.5$ **and** $\beta = 1.25$.

| n | $B_\alpha$ | $B_\beta$ | $M_\alpha$ | $M_\beta$ | $CI_\alpha$ | $CI_\beta$ |
|---|---|---|---|---|---|---|
| 20 | 0.1138 | 0.1161 | 0.1915 | 0.0873 | (0.9644 2.6549) | (0.9323 1.9691) |
| 45 | 0.0467 | 0.0441 | 0.0691 | 0.0269 | (1.1255 2.1280) | (1.0226 1.6626) |
| 70 | 0.0320 | 0.0291 | 0.0363 | 0.0154 | (1.1996 1.9391) | (1.0662 1.5397) |
| 95 | 0.0243 | 0.0172 | 0.0261 | 0.0102 | (1.2535 1.8551) | (1.0808 1.4767) |
| 120 | 0.0234 | 0.0171 | 0.0210 | 0.0081 | (1.2703 1.8207) | (1.1136 1.4701) |
| 145 | 0.0169 | 0.0105 | 0.0166 | 0.0071 | (1.2906 1.7761) | (1.1067 1.4310) |
| 170 | 0.0149 | 0.0087 | 0.0148 | 0.0056 | (1.3053 1.7556) | (1.1190 1.3980) |
| 195 | 0.0103 | 0.0042 | 0.0127 | 0.0050 | (1.3081 1.7417) | (1.1158 1.4012) |

**Table 4. Bias, MSE, and CI for MLE estimates** $\alpha = 1.75$ **and** $\beta = 1.5$.

| n | $B_\alpha$ | $B_\beta$ | $M_\alpha$ | $M_\beta$ | $CI_\alpha$ | $CI_\beta$ |
|---|---|---|---|---|---|---|
| 20 | 0.1576 | 0.1015 | 0.3065 | 0.0994 | (1.2111 3.2399) | (1.1158 2.2888) |
| 45 | 0.0662 | 0.0540 | 0.0924 | 0.0393 | (1.3262 2.5063) | (1.2418 1.9706) |
| 70 | 0.0358 | 0.0321 | 0.0554 | 0.0227 | (1.3646 2.3093) | (1.2849 1.8396) |
| 95 | 0.0240 | 0.0139 | 0.0388 | 0.0158 | (1.4303 2.2057) | (1.2960 1.7812) |
| 120 | 0.0248 | 0.0173 | 0.0291 | 0.0112 | (1.4884 2.1557) | (1.3195 1.7354) |
| 145 | 0.0195 | 0.0175 | 0.0235 | 0.0108 | (1.4899 2.0979) | (1.3458 1.7424) |
| 170 | 0.0107 | 0.0137 | 0.0201 | 0.0079 | (1.4988 2.0522) | (1.3524 1.7038) |
| 195 | 0.0135 | 0.0111 | 0.0177 | 0.0078 | (1.5419 2.0540) | (1.3478 1.6851) |

## NBUW distribution based on its quantiles

Let $\mu = Q_q(\alpha, \beta)$ and solve it for shape parameter $\alpha$ we get

$$\alpha = -\left(\frac{1}{\mu} - 1\right)^{-\beta} \log(q).$$

**Table 5. Bias, MSE, and CI for MLE estimates $\alpha$ = 2.0 and $\beta$ = 1.75.**

| n | $B_\alpha$ | $B_\beta$ | $M_\alpha$ | $M_\beta$ | $CI_\alpha$ | $CI_\beta$ |
|---|---|---|---|---|---|---|
| 20 | 0.2092 | 0.1327 | 0.4251 | 0.1542 | (1.3370 3.7476) | (1.3314 2.7657) |
| 45 | 0.0642 | 0.0624 | 0.1224 | 0.0494 | (1.5347 2.8937) | (1.4511 2.2605) |
| 70 | 0.0489 | 0.0352 | 0.0686 | 0.0298 | (1.6067 2.6226) | (1.4780 2.1331) |
| 95 | 0.0372 | 0.0315 | 0.0462 | 0.0227 | (1.6752 2.5053) | (1.5176 2.0794) |
| 120 | 0.0315 | 0.0179 | 0.0403 | 0.0168 | (1.6855 2.4622) | (1.5433 2.0418) |
| 145 | 0.0258 | 0.0196 | 0.0323 | 0.0138 | (1.7147 2.4264) | (1.5528 2.0065) |
| 170 | 0.0185 | 0.016 | 0.0259 | 0.0117 | (1.7353 2.3527) | (1.5814 1.9915) |
| 195 | 0.0199 | 0.0106 | 0.0227 | 0.0101 | (1.7521 2.3581) | (1.5748 1.9615) |

**Table 6. Bias, MSE, and CI for MLE estimates $\alpha$ = 2.5 and $\beta$ = 3.25.**

| n | $B_\alpha$ | $B_\beta$ | $M_\alpha$ | $M_\beta$ | $CI_\alpha$ | $CI_\beta$ |
|---|---|---|---|---|---|---|
| 20 | 0.3315 | 0.256 | 0.8754 | 0.5448 | (1.7211 5.1072) | (2.4117 5.0917) |
| 45 | 0.1197 | 0.1213 | 0.2356 | 0.1873 | (1.8916 3.7765) | (2.6110 4.2471) |
| 70 | 0.0841 | 0.0877 | 0.1409 | 0.1103 | (2.0072 3.4241) | (2.7349 4.0234) |
| 95 | 0.0629 | 0.0511 | 0.0899 | 0.0741 | (2.0827 3.2036) | (2.8360 3.8818) |
| 120 | 0.0437 | 0.0453 | 0.0711 | 0.0593 | (2.1093 3.1308) | (2.8515 3.7963) |
| 145 | 0.036 | 0.0313 | 0.0557 | 0.0480 | (2.1502 3.0625) | (2.8950 3.7593) |
| 170 | 0.041 | 0.0322 | 0.0489 | 0.0419 | (2.1494 2.9801) | (2.9017 3.6938) |
| 195 | 0.0289 | 0.0277 | 0.0383 | 0.0360 | (2.1806 2.9188) | (2.9159 3.6598) |

Now we have redefined the PDF and CDF of the NBUW model by re-parameterizing, and can be expressed as

$$\pi(y;\beta,\mu) = \beta \left[ -\left(\frac{1}{\mu}-1\right)^{-\beta} \log(q) \right] \left(\frac{1-y}{y}\right)^{(\beta-1)} y^{-2} \exp\left[ \left(\frac{1}{\mu}-1\right)^{-\beta} \log(q) \left(\frac{1-y}{y}\right)^{\beta} \right]. \tag{12}$$

$$G(y;\beta,\mu) = \exp\left[ \left(\frac{1}{\mu}-1\right)^{-\beta} \log(q) \left(\frac{1-y}{y}\right)^{\beta} \right], \tag{13}$$

where $q$ represents the quantile parameter and it is supposed to be known, $0 < y \leq 1$, $0 < \mu \leq 1$, and $\alpha > 0$. We have displayed the various shapes of Eq (12) in Fig (3). Quantile function based on parameterized CDF is given by;

$$Q(y;u) = \left[ \left[ \frac{\log(u)}{\log(q)} \left(\frac{1}{\mu}-1\right)^{\beta} \right]^{\frac{1}{\beta}} + 1 \right]^{-1}, \tag{14}$$

where $u$ has a uniform distribution over the interval (0,1). This quantile function is used to generate random samples from the NBUW distribution for specified quantiles.

## NBUW quantile regression model

Regression analysis is one of the most widely used statistical tools for understanding the relationship between a response variable and a set of covariates. The choice of regression model, however, often depends on the nature of the response variable. When the response variable is continuous and bounded within the unit interval, beta regression is a commonly employed approach as it models the conditional mean of the response variable effectively.

However, in situations where the response variable exhibits skewness or the presence of outliers, methods based on conditional means can be less robust. In such cases, quantile

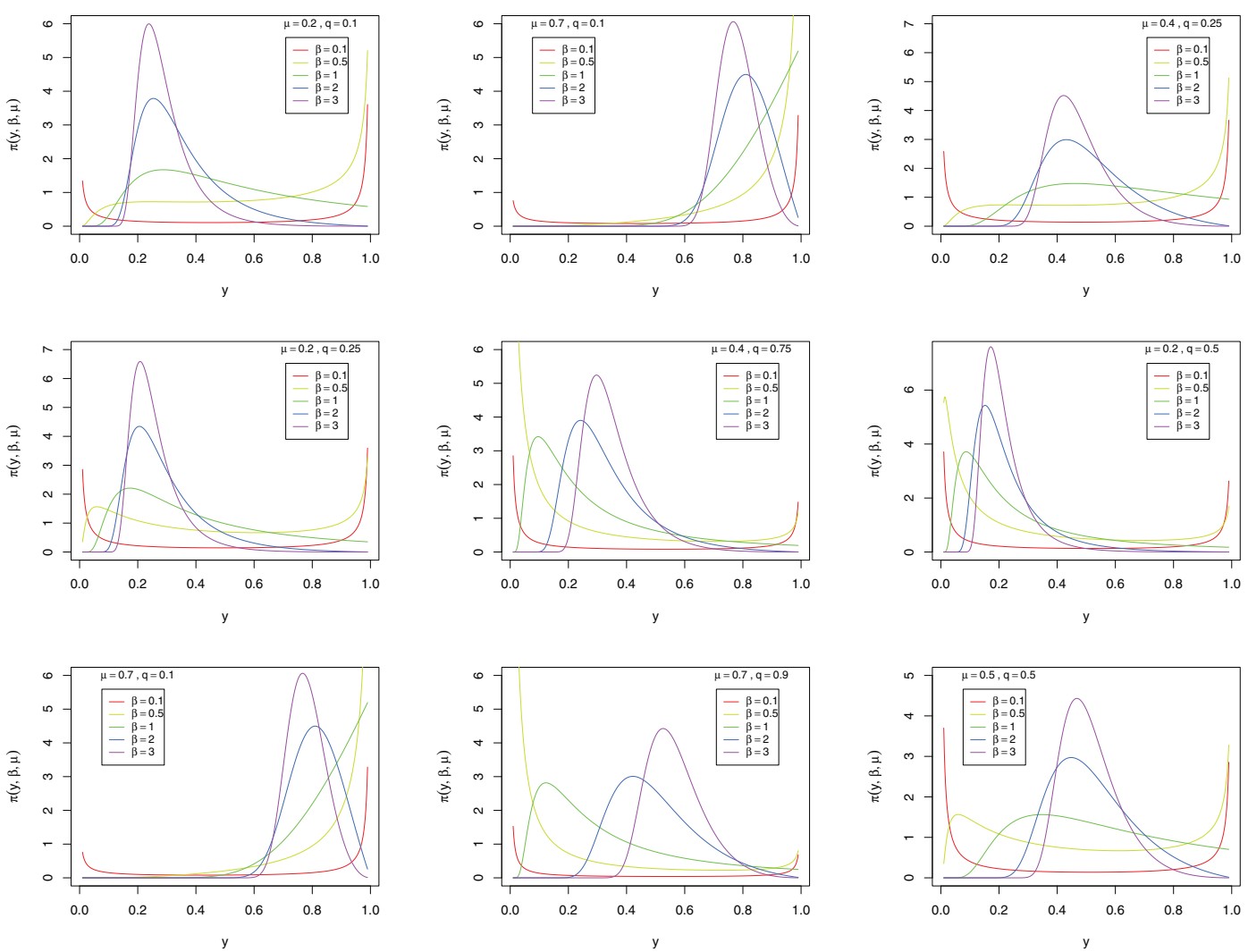

**Fig 3. Different shapes of PDF of the re-parameterized NBUW model.**

regression offers a more flexible alternative by directly modeling conditional quantiles of the response variable. Among unit interval quantile regression methods, the Kumaraswamy quantile regression model [3] is a popular choice. There is a constant need to improve existing models, either by refining their structure or introducing alternative approaches.

In this study, we propose a novel bounded unit Weibull quantile regression model that enhances the flexibility and applicability of unit interval quantile regression methods. Based on the density function described in Eq (12), the quantile regression model is outlined below.

Let $(y_1 \dots y_n)$ such that $y_i$ is an observation of $Y_i \sim NBUW(\beta, \mu_i, q)$ for $i = 1 \dots n$ with unknown parameters $\mu_i$ and $\beta$. The regression model based on quantile is defined as;

$$\pi(\mu_i) = \log\left(\frac{\mu_i}{1 - \mu_i}\right) = \mathbf{x_i}\delta^T, \tag{15}$$

where $\mathbf{x_i}$ is the $i^{th}$ design vector of predictors corresponding to the vector of regression coefficient $\delta$. The function $\pi(.)$ serves as the link function utilized to connect the linear predictors to the conditional quantile of the outcome variable. Further elaboration can be found in [36]. For example, when the parameter $q = 0.5$, quantile regression establishes the relationship between predictors and the median of the outcome variable. The logit link function is utilized to link the linear predictors to the conditional quantile of the response variable.

## Parameter estimation

The parameter estimation of the unit exponential quantile regression is determined using the maximum likelihood estimation method. After simplifying the regression equation presented in Eq (14), the following expression (see, Eq (16) for $\mu_i$ is obtained:

$$\mu_i = \frac{exp(x_i\delta^T)}{1 + exp(x_i\delta^T)}. \tag{16}$$

The log-likelihood function, used to estimate the unknown parameters of the NBUW quantile regression model, is defined as follows.

$$l(\Psi) = n\log(\beta) + \sum_{i=1}^{n}\log\left[-\left(\frac{1}{\mu_i} - 1\right)^{-\beta}\log(q)\right] + (\beta - 1)\sum_{i=1}^{n}\log\left(\frac{1 - y_i}{y_i}\right)$$
$$- 2\sum_{i=1}^{n}\log(y_i) + \sum_{i=1}^{n}\left[\left(\frac{1}{\mu_i} - 1\right)^{-\beta}\log(q)\left(\frac{1 - y_i}{y_i}\right)\right]. \tag{17}$$

where the unknown parameter vector is denoted as $\Psi = (\beta, \delta)^T$. To find the MLEs of $\Psi$, denoted $\hat{\Psi} = (\hat{\beta}, \hat{\delta})^T$, we maximize the likelihood function $l(\Psi)$ with respect to $\Psi$. The nonlinear nature of Eq (17) with respect to model parameters allows for direct maximization using software such as Matlab, R, Mathematica, Python, etc.

In particular, when $q = 0.5$, it represents modeling the conditional median. For maximization of Eq (17), we employ the maxLik package [35] of the R software. This package not only maximizes the likelihood function but also provides numerically computed asymptotic standard errors using the observed information matrix.

## Residual analysis

Residual analysis is required to evaluate the fit of the model. Randomized quantile residual is proposed for NBUW quantile regression. Detailed information on the randomized quantile residual can be found in [37]. The $i^{th}$ randomized quantile residual was obtained by using the following relations;

$$\hat{r}_i = \Phi^{-1}\left[G(y_i, \hat{\beta}, \hat{\mu}_i)\right],$$

where $G(y, \beta, \mu)$ represents the CDF of the re-parameterized NBUW model (referenced by Eq (12)), $\Phi^{-1}$ is the quantile function of the standard normal distribution, and $\hat{\mu}_i$ is estimated value of the quantile of the response variable. For a properly fitted model, the distribution of randomized quantile residuals should be standard normal.

Cox-Snell residuals are another form of residuals derived from the following relationship;

$$\hat{e}_i = -\log\left[1 - G(y_i, \hat{\beta}, \hat{\mu}_i)\right].$$

Detailed information on Cox-Snell residuals can be found in [38]. When the model fits the data appropriately, these residuals should follow an exponential distribution with a scale parameter of 1.0.

## Simulation

This section presents simulation studies employing five distinct simulation schemes to assess the parameter recovery of the NBUW quantile regression model. Each scheme is accompanied by the presentation of the relative percentage bias (RB) and the root mean square error (RMSE). All simulations were performed in R using the optim function. Comparable results were attained using the maxlik function. Below is the NBUW quantile regression model whose parameters were recovered using simulation:

$$logit(\mu_i) = \delta_0 + \delta_1 x_{i1}, i = 1, ..... n, \qquad (18)$$

where true value of parameters $\delta_0$ and $\delta_1$, and shape parameter $\beta$ are given below;

1. Simulation scheme 1: $\delta_0 = 1.3$, $\delta_1 = 1.4$ and $\beta = 2.0$
2. Simulation scheme 2: $\delta_0 = -2.0$, $\delta_1 = 1.0$ and $\beta = 1.5$
3. Simulation scheme 3: $\delta_0 = -0.2$, $\delta_1 = -0.6$ and $\beta = 2.5$
4. Simulation scheme 4: $\delta_0 = 0.7$, $\delta_1 = 0.4$ and $\beta = 3.0$
5. Simulation scheme 5: $\delta_0 = 1.0$, $\delta_1 = 2.0$ and $\beta = 4.0$

For each scheme, covariates were generated from a normal distribution with varying sample sizes: $n = 25, 75, 150$, and $300$, respectively. For each scheme, the Monte Carlo simulation was repeated 10,000 times. Each simulation scheme was repeated for $q = 0.1$, $q = 0.5$, and $q = 0.9$.

The simulation results for q = 0.5 were presented in Table 7. Relative absolute bias and RMSE were presented in Figs 4, 5, and 6. Relative bias and RMSE decrease as the sample size increases. For a sample size of 300, both bias and RMSE are close to 0, indicating an outstanding recovery of the parameters. Although the shape parameter exhibits a large relative bias for small sample sizes, it reduces significantly as the sample size increases. Likewise simulation results for $q = 0.1$ and $q = 0.9$ was also presented in Table 7 and Figs 7, 8, 9, 10 11, 12, respectively. All results consistently demonstrated satisfactory recovery of the proposed parameters at quantiles $q = 0.1$ and $q = 0.9$, highlighting the model's robustness and reliability.

## Application

### Educational attainment data

The educational attainment data, sourced from OECD.stats as cited by [21], encompass educational attainment percentages from 35 OECD countries, supplemented by four non-OECD countries. The dependent variable, the percentage of educational attainment, falls within the continuous range of 0 to 1. Educational attainment refers to the proportion of adults aged 25 to 64 years of age holding at least an upper secondary degree within the same age group population. In this examination, the covariates under consideration are life satisfaction and homicide rates. The distribution of the response variable, student attainment, skewed left with noticeable outliers (refer to Fig 13). Consequently, quantile regression, such as the model presented herein, proves suitable for modeling such data distributions. Therefore, the interplay between student attainment and predictors- namely, life satisfaction and homicide rates were explored utilizing the proposed new bounded unit Weibull quantile regression model.

**Table 7. Simulation results.**

| Scheme | $\delta_0$ | $\delta_1$ | $\beta$ | Sample | RB of $\delta_0$ | RB of $\delta_1$ | RB of $\beta$ | RMSE of $\delta_0$ | RMSE of $\delta_1$ | RMSE of $\beta$ | q |
|---|---|---|---|---|---|---|---|---|---|---|---|
| Scheme1 | 1.3 | 1.4 | 2.0 | 25 | 2.8159 | -0.3587 | 7.3744 | 0.1140 | 0.0787 | 0.2218 | 0.1 |
| Scheme1 | 1.3 | 1.4 | 2.0 | 75 | 0.9740 | -0.0334 | 2.3976 | 0.0631 | 0.0682 | 0.1041 | 0.1 |
| Scheme1 | 1.3 | 1.4 | 2.0 | 150 | 0.4582 | -0.0066 | 1.0995 | 0.0430 | 0.0427 | 0.0684 | 0.1 |
| Scheme1 | 1.3 | 1.4 | 2.0 | 300 | 0.2267 | -0.0136 | 0.5745 | 0.0307 | 0.0306 | 0.0469 | 0.1 |
| Scheme2 | -2.0 | 1.0 | 1.5 | 25 | -2.4405 | -0.6696 | 7.3744 | 0.1520 | 0.1049 | 0.1992 | 0.1 |
| Scheme2 | -2.0 | 1.0 | 1.5 | 75 | -0.8441 | -0.0624 | 2.3976 | 0.0842 | 0.0909 | 0.0991 | 0.1 |
| Scheme2 | -2.0 | 1.0 | 1.5 | 150 | -0.3971 | -0.0124 | 1.0995 | 0.0573 | 0.0569 | 0.0668 | 0.1 |
| Scheme2 | -2.0 | 1.0 | 1.5 | 300 | -0.1965 | -0.0253 | 0.5745 | 0.0409 | 0.0407 | 0.0463 | 0.1 |
| Scheme3 | 0.7 | 0.4 | 3.0 | 25 | 3.4863 | -0.8369 | 7.3744 | 0.0760 | 0.0525 | 0.2764 | 0.1 |
| Scheme3 | 0.7 | 0.4 | 3.0 | 75 | 1.2059 | -0.0780 | 2.3976 | 0.0421 | 0.0455 | 0.1171 | 0.1 |
| Scheme3 | 0.7 | 0.4 | 3.0 | 150 | 0.5673 | -0.0154 | 1.0995 | 0.0287 | 0.0285 | 0.0726 | 0.1 |
| Scheme3 | 0.7 | 0.4 | 3.0 | 300 | 0.2807 | -0.0317 | 0.5745 | 0.0205 | 0.0204 | 0.0486 | 0.1 |
| Scheme4 | 1.0 | 2.0 | 4.0 | 25 | 1.8303 | -0.1255 | 7.3743 | 0.0570 | 0.0393 | 0.3383 | 0.1 |
| Scheme4 | 1.0 | 2.0 | 4.0 | 75 | 0.6331 | -0.0117 | 2.3976 | 0.0316 | 0.0341 | 0.1332 | 0.1 |
| Scheme4 | 1.0 | 2.0 | 4.0 | 150 | 0.2978 | -0.0023 | 1.0995 | 0.0215 | 0.0213 | 0.0783 | 0.1 |
| Scheme4 | 1.0 | 2.0 | 4.0 | 300 | 0.1473 | -0.0048 | 0.5744 | 0.0153 | 0.0153 | 0.0510 | 0.1 |
| Scheme1 | 1.3 | 1.4 | 2.0 | 25 | 0.2311 | -0.3587 | 7.2744 | 0.119 | 0.0787 | 0.2218 | 0.5 |
| Scheme1 | 1.3 | 1.4 | 2.0 | 75 | 0.0848 | 0.0334 | 2.3976 | 0.0678 | 0.0682 | 0.1041 | 0.5 |
| Scheme1 | 1.3 | 1.4 | 2.0 | 150 | 0.0517 | -0.0066 | 1.0995 | 0.0485 | 0.0427 | 0.0684 | 0.5 |
| Scheme1 | 1.3 | 1.4 | 2.0 | 300 | 0.0105 | -0.0136 | 0.5745 | 0.0344 | 0.0306 | 0.0469 | 0.5 |
| Scheme2 | -2.0 | 1.0 | 1.5 | 25 | -0.3011 | -0.989 | 7.3824 | 0.2379 | 0.173 | 0.1814 | 0.5 |
| Scheme2 | -2.0 | 1.0 | 1.5 | 75 | -0.0914 | -0.3898 | 2.3931 | 0.1372 | 0.1143 | 0.0961 | 0.5 |
| Scheme2 | -2.0 | 1.0 | 1.5 | 150 | -0.0689 | -0.0678 | 1.1016 | 0.0964 | 0.0796 | 0.0656 | 0.5 |
| Scheme2 | -2.0 | 1.0 | 1.5 | 300 | -0.0151 | -0.1308 | 0.5723 | 0.0688 | 0.0582 | 0.0456 | 0.5 |
| Scheme3 | 0.7 | 0.4 | 3.0 | 25 | 0.6604 | -1.9315 | 7.3744 | 0.183 | 0.1211 | 0.1914 | 0.5 |
| Scheme3 | 0.7 | 0.4 | 3.0 | 75 | 0.2514 | -0.2121 | 2.4028 | 0.1044 | 0.1049 | 0.0975 | 0.5 |
| Scheme3 | 0.7 | 0.4 | 3.0 | 150 | 0.131 | -0.4455 | 1.1115 | 0.0747 | 0.0596 | 0.0664 | 0.5 |
| Scheme3 | 0.7 | 0.4 | 3.0 | 300 | 0.0325 | -0.112 | 0.5663 | 0.0527 | 0.0437 | 0.0463 | 0.5 |
| Scheme4 | 1.0 | 2.0 | 4.0 | 25 | 0.1502 | -0.1255 | 7.3745 | 0.0595 | 0.0393 | 0.3383 | 0.5 |
| Scheme4 | 1.0 | 2.0 | 4.0 | 75 | 0.0551 | -0.0117 | 2.3977 | 0.0339 | 0.0341 | 0.1332 | 0.5 |
| Scheme4 | 1.0 | 2.0 | 4.0 | 150 | 0.0335 | -0.0023 | 1.0996 | 0.0243 | 0.0213 | 0.0783 | 0.5 |
| Scheme4 | 1.0 | 2.0 | 4.0 | 300 | 0.0067 | -0.0047 | 0.5746 | 0.0172 | 0.0153 | 0.051 | 0.5 |
| Scheme1 | 1.3 | 1.4 | 2.0 | 25 | -3.8245 | -0.3587 | 7.3741 | 0.2348 | 0.0787 | 0.2218 | 0.9 |
| Scheme1 | 1.3 | 1.4 | 2.0 | 75 | -1.3101 | -0.0334 | 2.3973 | 0.1340 | 0.0682 | 0.1041 | 0.9 |
| Scheme1 | 1.3 | 1.4 | 2.0 | 150 | -0.5860 | -0.0066 | 1.0992 | 0.0961 | 0.0427 | 0.0684 | 0.9 |
| Scheme1 | 1.3 | 1.4 | 2.0 | 300 | -0.3286 | -0.0136 | 0.5742 | 0.0676 | 0.0306 | 0.0469 | 0.9 |
| Scheme2 | -2.0 | 1.0 | 1.5 | 25 | 3.3145 | -0.6696 | 7.3740 | 0.3131 | 0.1049 | 0.1992 | 0.9 |
| Scheme2 | -2.0 | 1.0 | 1.5 | 75 | 1.1354 | -0.0624 | 2.3973 | 0.1786 | 0.0909 | 0.0991 | 0.9 |
| Scheme2 | -2.0 | 1.0 | 1.5 | 150 | 0.5078 | -0.0124 | 1.0992 | 0.1281 | 0.0569 | 0.0668 | 0.9 |
| Scheme2 | -2.0 | 1.0 | 1.5 | 300 | 0.2847 | -0.0253 | 0.5741 | 0.0901 | 0.0407 | 0.0463 | 0.9 |
| Scheme3 | 0.7 | 0.4 | 3.0 | 25 | -4.7354 | -0.8369 | 7.3743 | 0.1566 | 0.0525 | 0.2764 | 0.9 |
| Scheme3 | 0.7 | 0.4 | 3.0 | 75 | -1.6223 | -0.0780 | 2.3975 | 0.0893 | 0.0455 | 0.1171 | 0.9 |
| Scheme3 | 0.7 | 0.4 | 3.0 | 150 | -0.7256 | -0.0154 | 1.0993 | 0.0640 | 0.0285 | 0.0726 | 0.9 |
| Scheme3 | 0.7 | 0.4 | 3.0 | 300 | -0.4070 | -0.0317 | 0.5743 | 0.0450 | 0.0204 | 0.0486 | 0.9 |
| Scheme4 | 1.0 | 2.0 | 4.0 | 25 | -2.4862 | -0.1255 | 7.3745 | 0.1174 | 0.0393 | 0.3383 | 0.9 |
| Scheme4 | 1.0 | 2.0 | 4.0 | 75 | -0.8518 | -0.0117 | 2.3976 | 0.0670 | 0.0341 | 0.1332 | 0.9 |
| Scheme4 | 1.0 | 2.0 | 4.0 | 150 | -0.3811 | -0.0023 | 1.0995 | 0.0480 | 0.0213 | 0.0783 | 0.9 |
| Scheme4 | 1.0 | 2.0 | 4.0 | 300 | -0.2138 | -0.0047 | 0.5745 | 0.0338 | 0.0153 | 0.0510 | 0.9 |

We compared three popular regression models with the NBUW quantile regression model. These models include beta regression [13], Kumaraswamy quantile regression [3], and unit-Weibull quantile regression [39]. In this applied example, the following regression model was

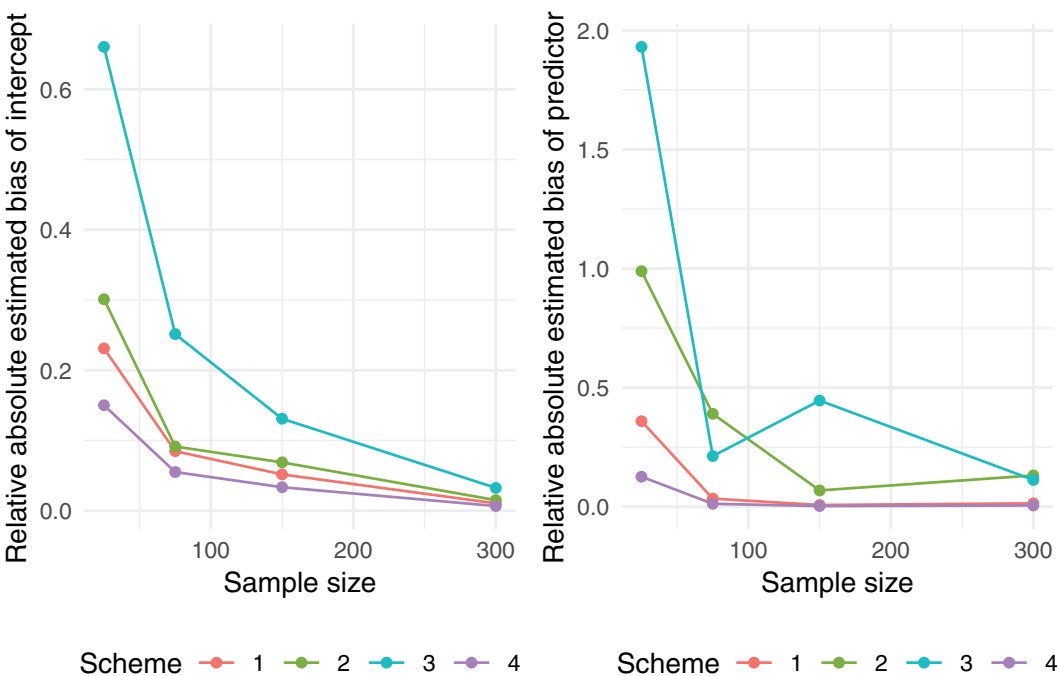

**Fig 4. Relative absolute bias of intercept and predictor for q = 0.5.**

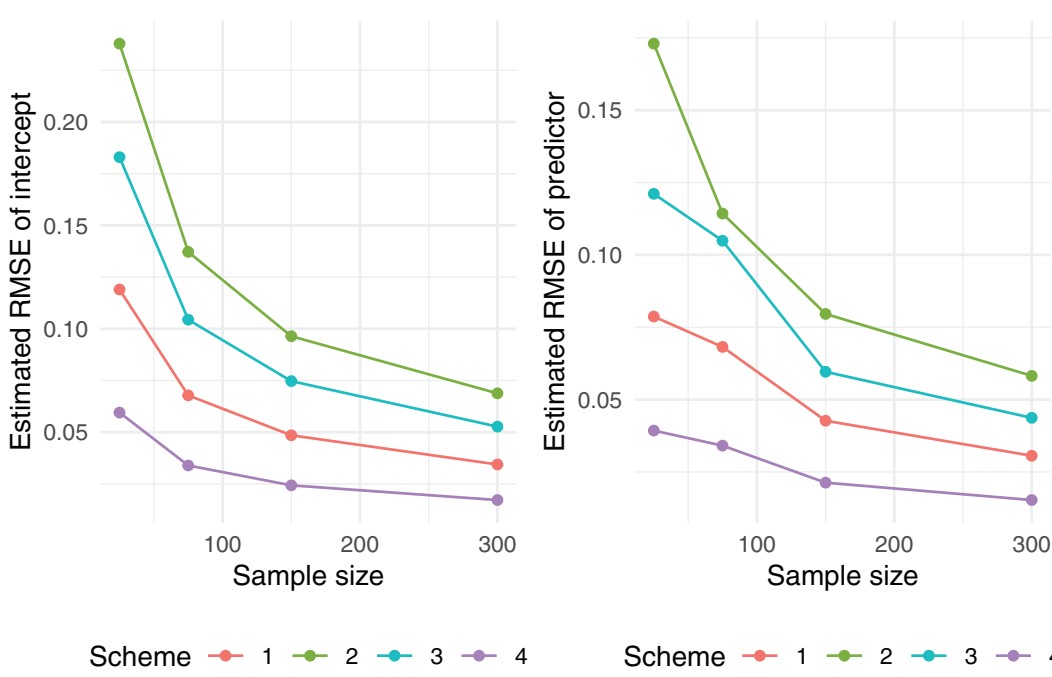

**Fig 5. RMSE of regression coefficients of intercept and predictor for q = 0.5.**

used.

$$logit(\mu_i) = \delta_0 + \delta_1 z_{i1} + \delta_2 z_{i2}, i = 1, .....41, \tag{19}$$

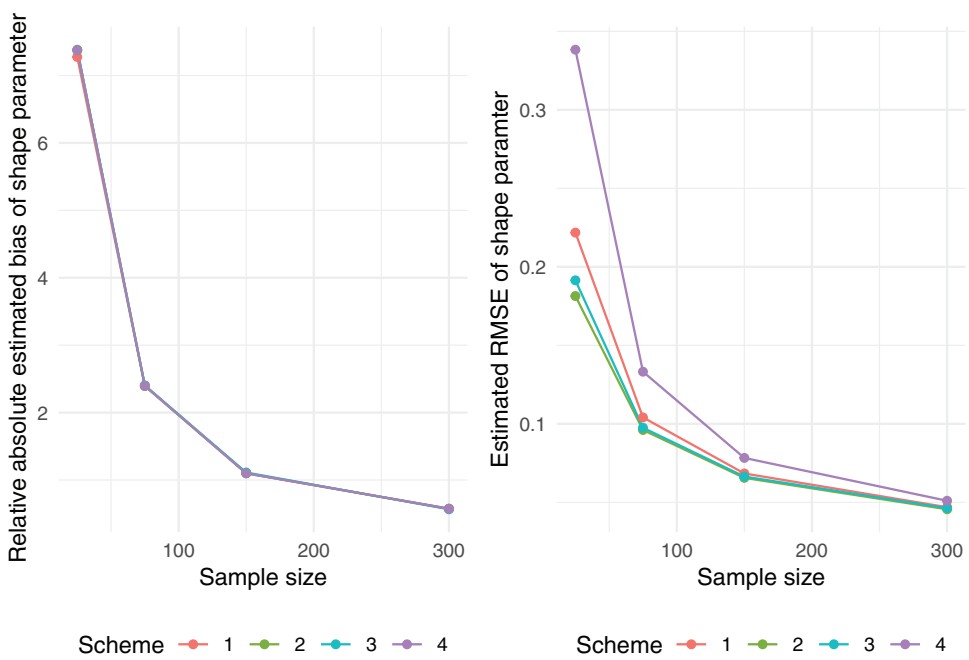

**Fig 6. Relative absolute bias and RMSE of estimated shape parameter for q = 0.5.**

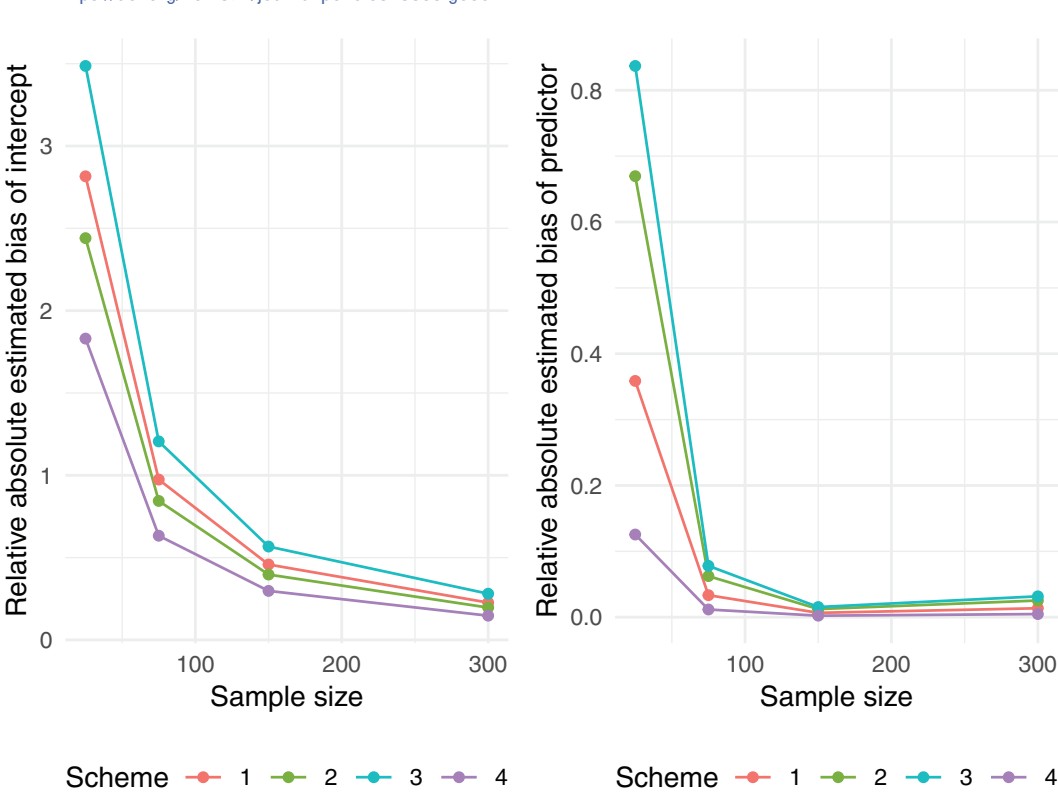

**Fig 7. Relative absolute bias of intercept and predictor for q = 0.1.**

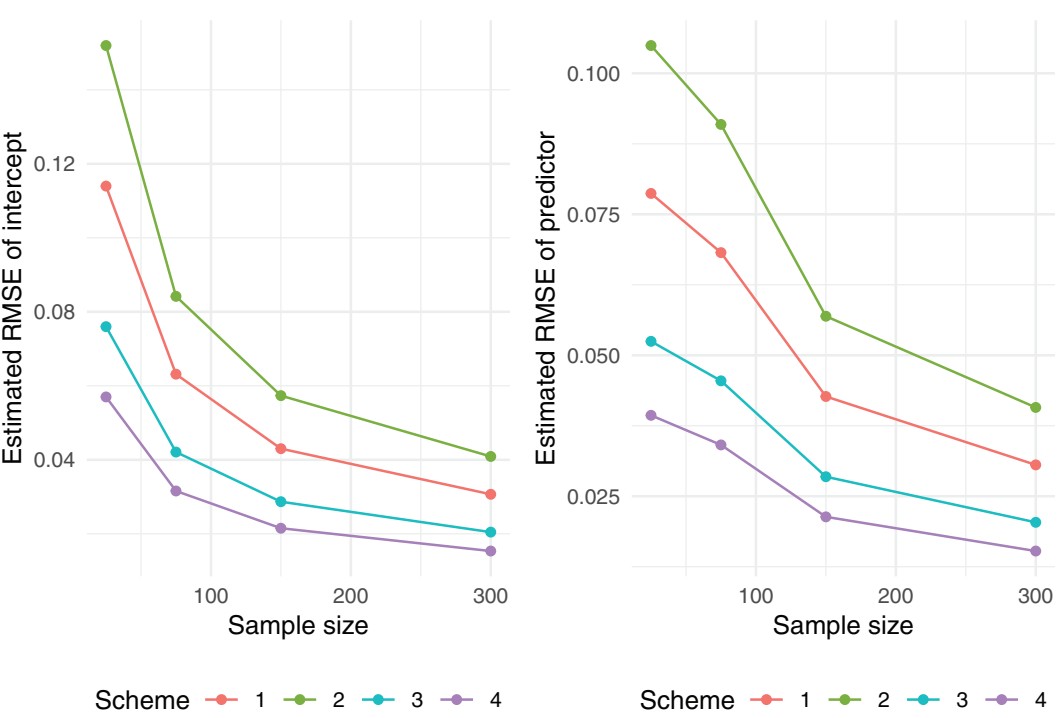

**Fig 8. RMSE of regression coefficients of intercept and predictor for q = 0.1.**

**Fig 9. Relative absolute bias and RMSE of estimated shape parameter for q = 0.1.**

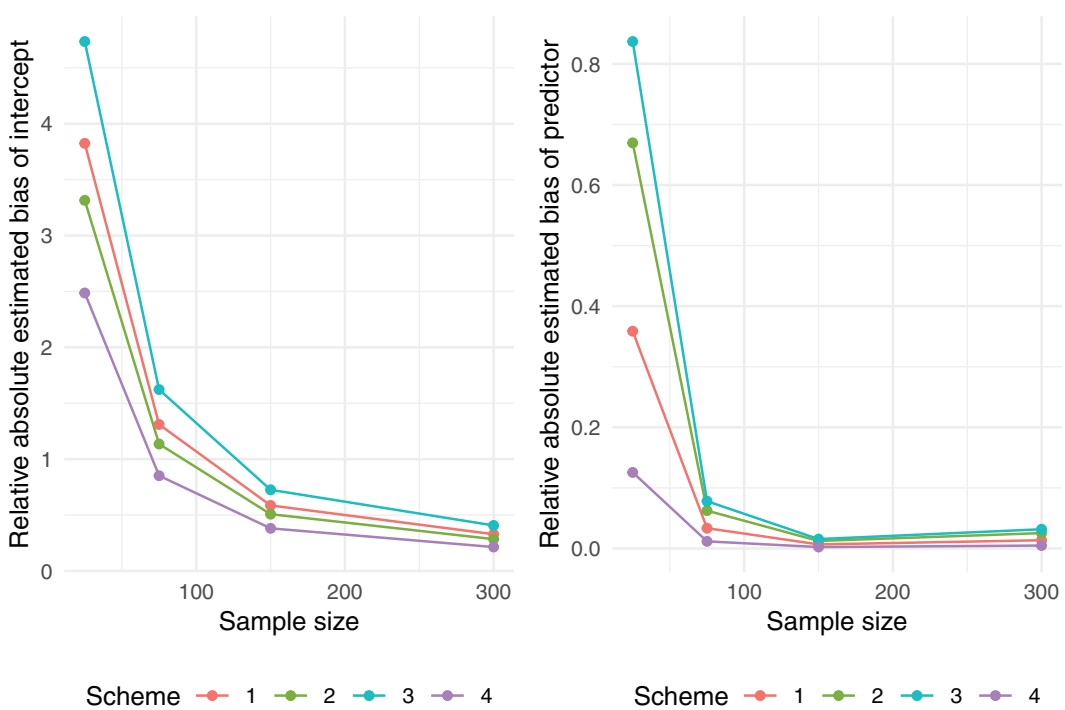

**Fig 10. Relative absolute bias of intercept and predictor for q = 0.9.**

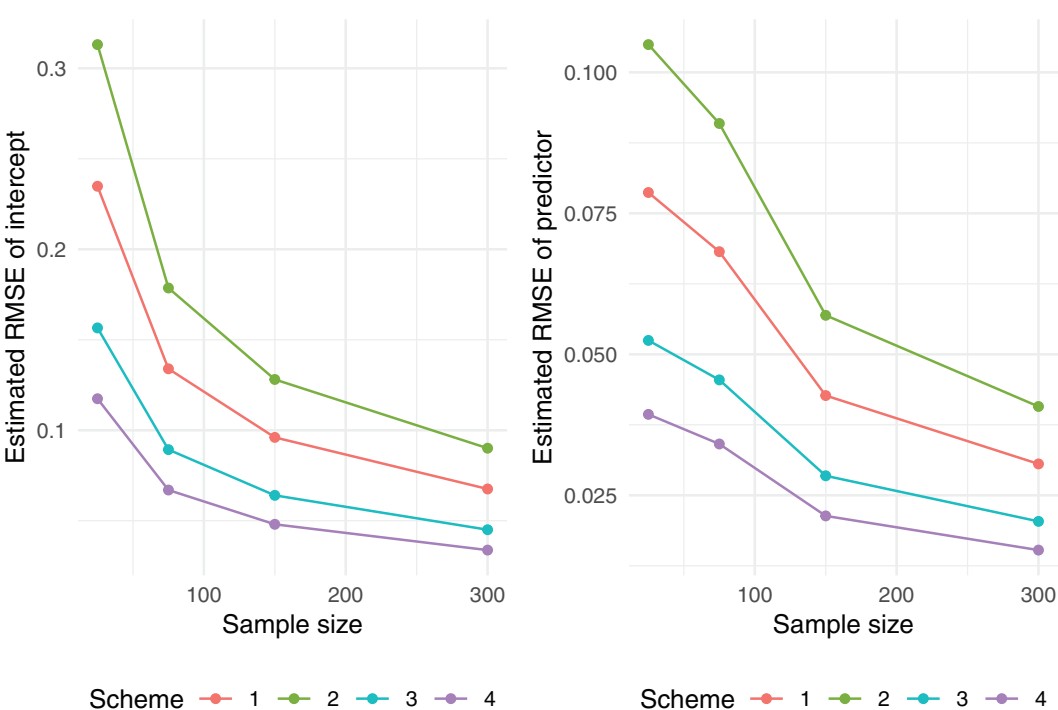

**Fig 11. RMSE of regression coefficients of intercept and predictor for q = 0.9**

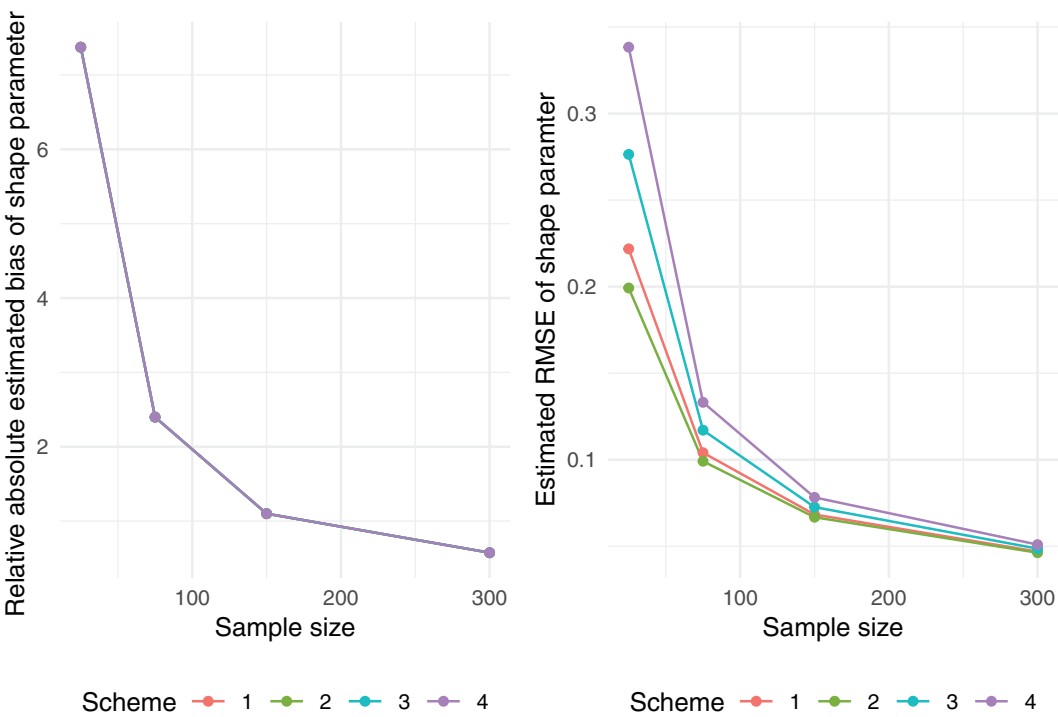

**Fig 12. Relative absolute bias and RMSE of estimated shape parameter for q = 0.9.**

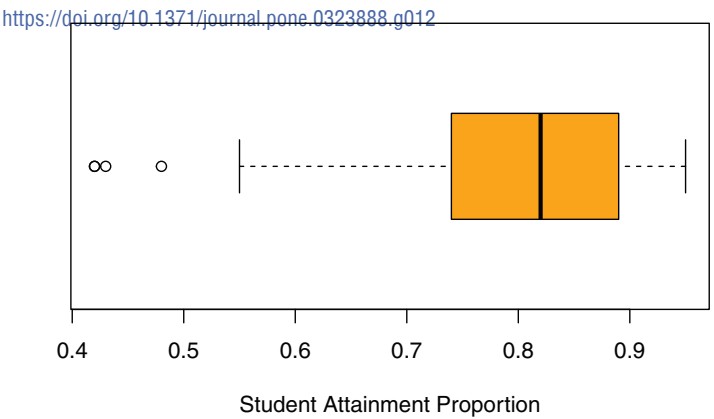

**Fig 13. Boxplot of student attainment proportions.**

where $\mu_i$ is the median for Kumaraswamy, unit-Weibull, and NBUW, and the mean for the beta regression model. $z_1$ is life satisfaction, and $z_2$ is homicide. Parameter estimation was conducted using the MLE method. The `unitquantreg` package in R [40] was utilized for estimating the parameters of the Kumaraswamy and unit-Weibull regression model. For the beta regression model, parameter estimation was facilitated by the `betareg` package [41]. Custom code employing the maxLik function from the maxLik package in R was employed for estimating the parameters of the NBUW regression model. Model performance assessment comprised generating QQ plots of randomized quantile residuals and applying the Kolmogorov-Smirnov test to evaluate model suitability. Additionally, comparative analysis across different models was conducted using the AIC and BIC.

The results of the parameter estimation are presented in Table 8. The NBUW regression model exhibited superior performance based on the AIC and BIC metrics compared to the Kumaraswamy and beta regression models. In addition, the AIC and BIC values of the Unit-Weibull and NBUW models were found to be comparable. The K-S test results affirmed satisfactory model fit across all models. Fig 14 illustrates a QQ plot of quantile residuals. It also confirms the superior performance of the NBUW quantile regression.

Furthermore, the effectiveness of the proposed quantile regression model was evaluated against the unit logistic, unit Johnson, and unit Burr XII models. The AIC and BIC values for the Unit Burr XII model were -66.17 and -59.31 respectively, for the unit logistic quantile

**Table 8. Parameter estimation and model comparison statistics.**

| Parameter | Kumarawwami | | Beta | | Unit Weibull | | NBUW | |
|---|---|---|---|---|---|---|---|---|
| | Estimate | *p*-value | Estimate | *p*-value | Estimate | *p*-value | Estimate | *p*-value |
| Intercept | 1.6770 | 0.1680 | -0.1760 | 0.8670 | -1.6470 | 0.0760 | -2.0250 | 0.0323 |
| Satisfaction | -0.0014 | 0.9950 | 0.2530 | 0.1053 | 0.4780 | 0.0003 | 0.5323 | 0.0001 |
| Homicide | -0.0709 | 0.0000 | -0.0546 | 0.0007 | -0.04312 | 0.0237 | -0.0438 | 0.0127 |
| $\beta$ | 1.9200 | 0.0000 | 12.0190 | 0.0000 | 0.6017 | 0.0000 | 1.5700 | 0.0002 |
| $\hat{l}$ | 34.5300 | | 35.0500 | | 37.3900 | | 37.1800 | |
| AIC | -61.0500 | | -62.0900 | | -66.7900 | | -66.3600 | |
| BIC | -54.2000 | | -55.2400 | | -59.9400 | | -59.5000 | |
| KS | 0.1395 | 0.3680 | 0.1418 | 0.3486 | 0.0980 | 0.7880 | 0.0944 | 0.8248 |

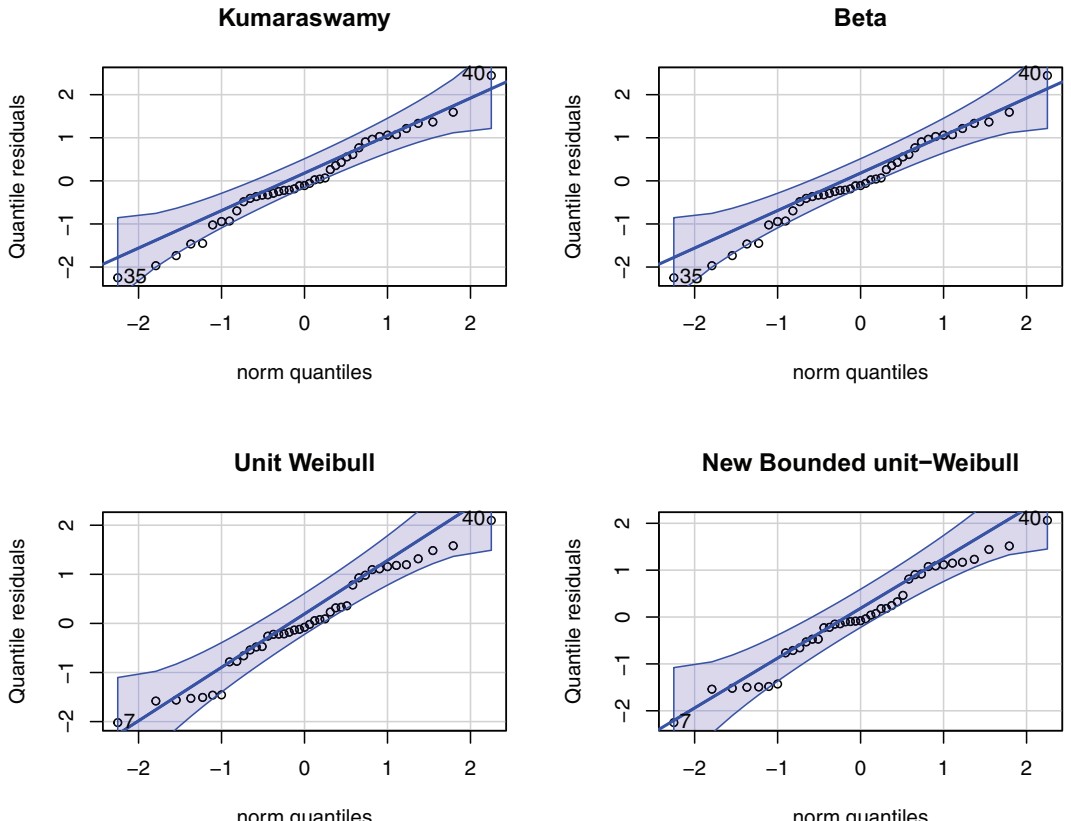

**Fig 14. QQ Plot of randomized quantile residuals.**

regression model they were -64.34 and -57.49, and for the unit Johnson quantile regression model, they were -65.54 and -58.68. In contrast, our proposed model yielded AIC and BIC values of -66.36 and -59.5, respectively. It also highlights the superiority of the NBUW model over existing models in the literature.

The results of the NBUW quantile regression analysis indicate that the percentage of student achievement is significantly associated with both life satisfaction and homicide. The results further show that student attainment is positively associated with life satisfaction and negatively associated with homicide. The results of both the UW and NBUW quantile regressions exhibit consistency. However, these findings diverge from those of the beta and Kumaraswamy regressions, where only homicide demonstrates a statistically significant negative relationship with student attainment. It is intriguing to observe that the fitting of the regression model, the direction of the association, and the significance of the predictor variables with the response variable all depend on the choice of distribution. In the current data scenario, the superior fit of NBUW quantile regression over other competing models is confirmed by metrics such as AIC, BIC, and the QQ plot of randomized residuals.

Hence, it is crucial to select the distribution that best fits the data at hand. In the presence of outliers, opting for quantile regression is prudent. Furthermore, our proposed model proves to be a favorable option for practitioners dealing with bounded response variables containing outliers.

## Risk assessment data

The data utilized in this study originate from the managerial cost-effectiveness investigation conducted by [30]. Furthermore, it can be accessed through the personal web page of Prof. E. Frees at https://sites.google.com/a/wisc.edu/jed-frees/. The purpose of gathering the data was to evaluate the cost-effectiveness of the company's management philosophy to mitigate its exposure to different property losses and accidents while considering the unique characteristics of the company. The data set consists of seven variables, with firm cost serving as the response variable and the remaining variables as covariates. Each variable is described below based on the information provided by the source:

- **Firm cost**($y$): This variable measures the evaluation of the effectiveness of the firm's risk management in terms of cost. It includes the total property and casualty premiums as well as uninsured losses, expressed as a percentage.
- **Assume**($z_1$): Denotes the per-occurrence retention amount as a percentage of total assets.
- **Cap**($z_2$): Indicates ownership of a captive insurance entity by the firm.
- **Size log**($z_3$): Natural logarithm of total assets.
- **Indcost**($z_4$): Represents the firm's industrial risk measure.
- **Central**($z_5$): A measure indicating the significance of local managers in determining the retained risk amount.
- **Soph**($z_6$): Indicates the degree of importance placed on utilizing analytical tools.

The response variable, firm cost, exhibits right-skewness along with the presence of outliers (see Fig 15). Consequently, our proposed quantile regression model is particularly suited to handle such data characteristics. To investigate the relationship between firm costs and the specified predictor variables, we compared three widely used regression models, beta, Kumaraswamy, and unit-Weibull, with the NBUW quantile regression model, utilizing risk assessment data. The formulation of the quantile regression model is as follows:

$$logit(\mu_i) = \delta_0 + \delta_1 z_{i1} + \delta_2 z_{i2} + \delta_3 z_{i3} + \delta_4 z_{i4} + \delta_5 z_{i5} + \delta_6 z_{i6}, i = 1, ..., 73, \tag{20}$$

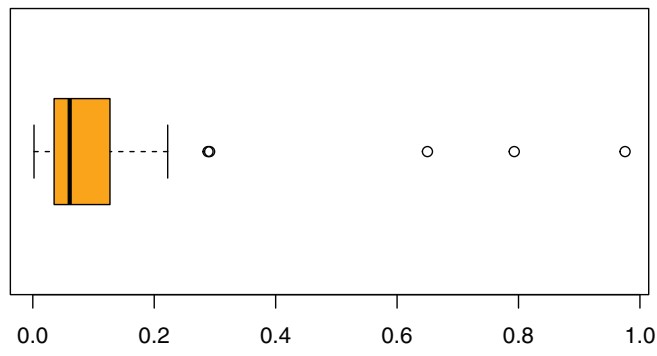

**Fig 15. Box-plot of firm cost.**

where $\mu_i$ represents the median for Kumaraswamy, unit-Weibull, and NBUW, and the mean for the beta regression model. $\delta_0, \delta_1, \delta_2, \delta_3, \delta_4, \delta_5, \delta_6$ are regression coefficients. The model parameters were estimated using a method similar to that outlined in the educational attainment data analysis. The performance of the model was evaluated through the plotting of half-normal quantile residual plots. The suitability of the model was evaluated using the Kolmogorov-Smirnov test, and model comparison was conducted using the AIC and BIC.

The parameter estimation results are detailed in Table 9. Based on the AIC and BIC criteria, the NBUW regression model outperformed the commonly used Kumaraswamy and beta regression models, as well as the unit-Weibull regression model. K-S test results indicated satisfactory model fits for all models except the beta regression model. The half-normal plot of quantile residuals, depicted in Fig 16, illustrates that the NBUW regression exhibits greater robustness to outliers compared to the mean-based beta regression and other competing models.

Furthermore, we compared our proposed quantile regression model with unit logistic quantile regression, unit Johnson quantile regression, and unit Burr XII quantile regression models. The AIC and BIC values for the Unit Burr II model were -87.02 and -68.69, respectively; for the unit logistic quantile regression model, they were -222.67 and -204.34, and for the unit Johnson quantile regression model, they were -200.18 and -181.85. In contrast, our proposed model yielded AIC and BIC values of -223.69 and -206.36, respectively.

**Table 9. Parameter estimation and model comparison statistics of risk assessment data.**

| Parameter | Kumarawwami | | Beta | | Unit Weibull | | NBUW | |
|---|---|---|---|---|---|---|---|---|
| | Estimate | *p*-value | Estimate | *p*-value | Estimate | *p*-value | Estimate | *p*-value |
| Intercept | 2.5380 | 0.1610 | 1.8880 | 0.1029 | 3.4720 | 0.0025 | 4.2976 | 0.0000 |
| Assume | -0.0364 | 0.0749 | -0.0121 | 0.3579 | -0.0076 | 0.5280 | -0.0041 | 0.7625 |
| Cap | 0.5962 | 0.6557 | 0.1779 | 0.4535 | 0.1278 | 0.6029 | -0.0063 | 0.9775 |
| Sizelog | -7981 | 0.0000 | -0.5114 | 0.0000 | -0.8043 | 0.0000 | -0.9342 | 0.0000 |
| Indcost | 5.2574 | 0.2079 | 1.2362 | 0.0104 | 1.4390 | 0.0006 | 1.7449 | 0.0000 |
| Central | -0.0278 | 0.8530 | -0.0122 | 0.8920 | -0.0241 | 0.7779 | -0.0433 | 0.5945 |
| Soph | -0.0274 | 0.5120 | -0.0037 | 0.8632 | -0.0023 | 0.9132 | 0.0005 | 0.9797 |
| $\beta$ | -0.0217 | 0.8350 | 6.3310 | 0.0000 | 1.2090 | 0.0000 | 1.1800 | 0.0551 |
| $\hat{l}$ | 98.8200 | | 87.7200 | | 111.1100 | | 119.8400 | |
| AIC | -181.6500 | | -159.4400 | | -206.2200 | | -223.6900 | |
| BIC | -163.3290 | | -141.1200 | | -187.8900 | | -205.3600 | |
| KS | 0.1196 | 0.2278 | 0.3358 | 0.0000 | 0.0779 | 0.7365 | 0.0849 | 0.6361 |

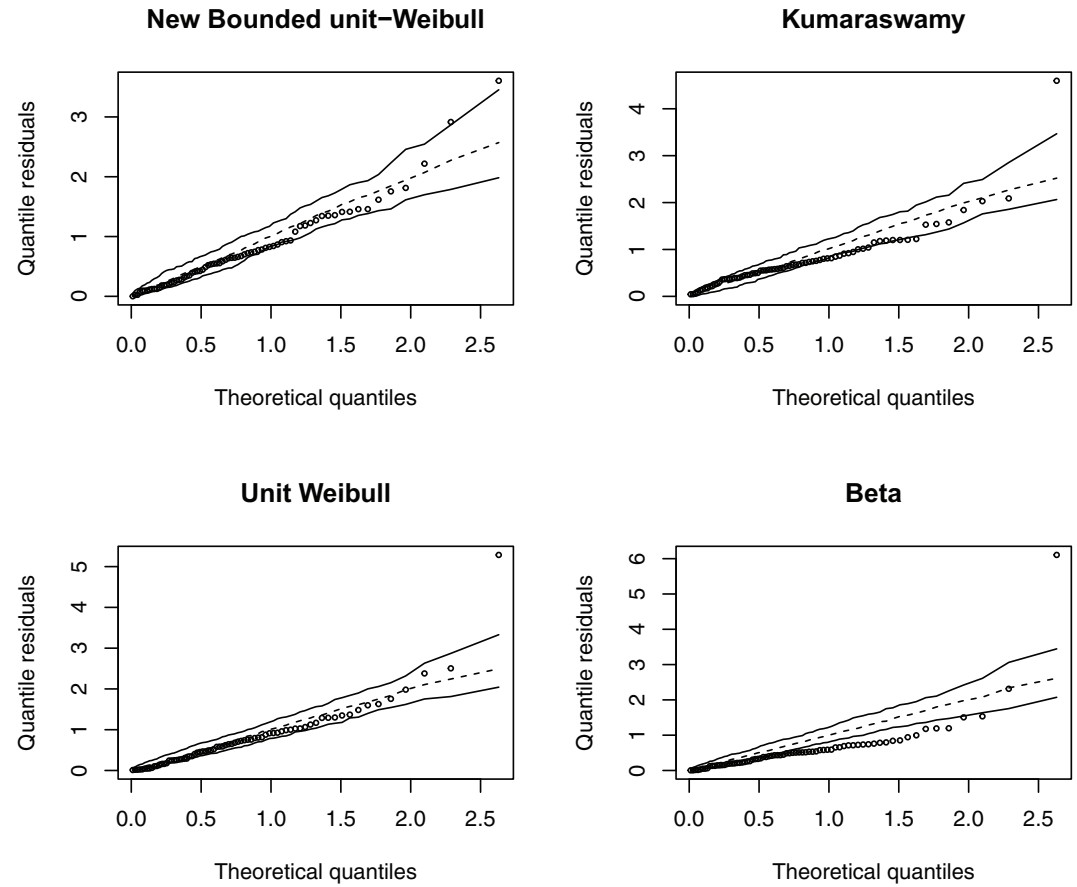

**Fig 16. Half normal plot of randomized quantile residuals.**

From the results derived from NBUW quantile regression, only the logarithm of firm size and the firm's industrial risk measure are statistically significant. The logarithm of the size of the firm is negatively associated, while the industrial risk measure is positively associated with the cost effectiveness of risk management of the firm. The remaining variables are not statistically significant. The results exhibit consistency across UW, NBUW, and beta regression models, except goodness of fit. However, in Kumaraswamy quantile regression, only the logarithm of firm size is significant, while other variables do not demonstrate statistical significance. It confirms the importance of selecting the appropriate distribution when analyzing the relationship between a bounded response variable and predictors. Several other studies have also reached conclusions similar to ours [19,21]. Furthermore, in situations such as risk assessment data where outliers are present, median-based regressions prove to be more robust. Upon examining the AIC, BIC, and residual plots, it becomes evident that our proposed NBUW distribution and its associated quantile regression are optimal choices for practitioners.

## Conclusion

In summary, this research introduces a new bounded unit Weibull distribution designed specifically for the (0,1) domain, which plays a critical role in characterizing phenomena in

various applied sciences. Our exploration has unveiled several intriguing properties, including different moments and their generating function, entropies, and a linear form of the beta distribution. In addition, we have devised the SPRT and ASN for the proposed model and established a new quantile regression model. A simulation study was carried out to evaluate the parameter recovery of the proposed new bounded unit Weibull distribution and the associated quantile regression model, which yielded satisfactory results for all parameters.

The proposed quantile regression model was also applied to two real datasets: one on student attainment and the other on risk assessment. For the risk assessment dataset, our proposed model outperformed the well-known beta and Kumaraswami regression models, as well as the UW regression model, based on two information criteria (AIC and BIC) and residual plots. Furthermore, the NBUW quantile regression model demonstrated superiority over log-logistic, Johnson, and Burr XII quantile regression models based on AIC and BIC. Similarly, for the education attainment dataset, the NBUW regression model outperformed the beta and Kumaraswamy regression models, while NBUW performed comparably to the UW quantile regression model. Furthermore, in the context of the educational attainment dataset, the NBUW quantile regression model exhibited superiority over the log-logistic, Johnson, and Burr XII quantile regression models based on AIC and BIC criteria. Based on the findings, when outliers are present in the dataset, such as in risk assessment data, median-based regression proves to be robust to these outliers. Specifically, NBUW quantile regression could be a favorable model choice for practitioners in such situations. Although our proposed model may not be applicable when zeros are present in the dataset, it could represent a superior option for researchers dealing with bounded data situations within the range of 0 and 1. Its superiority has been demonstrated in applied data situations presented in this paper. These findings underscore the importance of our contributions to the advancement of the statistical toolkit to analyze bounded variables in various scientific disciplines.

## Limitations

Although the NBUW model has several advantages over the existing models, it is not free from limitations. Below are some of the limitations and potential directions for future research:

1. In real-world applications, practitioners often require a model that can accommodate zeros. The NBUW model lacks this feature; therefore, future research should focus on adapting it to handle excess zeros too.
2. NBUW distribution is univariate; therefore, future research should focus on extending it to multivariate settings.
3. NBUW quantile regression requires independent responses and is not capable of modeling correlated outcomes. Therefore, future research should focus on extending it to mixed-effect models to properly accommodate correlated responses.

## Supporting information

**S1 Data. Datasets used in this study.**
(ZIP)

## Author contributions

**Conceptualization:** Laxmi Prasad Sapkota.

**Formal analysis:** Laxmi Prasad Sapkota, Nirajan Bam.

**Investigation:** Laxmi Prasad Sapkota.

**Methodology:** Laxmi Prasad Sapkota, Nirajan Bam, Vijay Kumar.

**Project administration:** Laxmi Prasad Sapkota.

**Resources:** Laxmi Prasad Sapkota.

**Software:** Laxmi Prasad Sapkota, Nirajan Bam, Vijay Kumar.

**Supervision:** Vijay Kumar.

**Writing – original draft:** Laxmi Prasad Sapkota, Nirajan Bam.

**Writing – review & editing:** Laxmi Prasad Sapkota, Nirajan Bam, Vijay Kumar.

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
