## [Decision Letter · Decision Letter 0]

7 Mar 2025

PONE-D-25-02426New Bounded Unit Weibull Model: Applications with Quantile RegressionPLOS ONE

Dear Dr. Sapkota,

Thank you for submitting your manuscript to PLOS ONE. After careful consideration, we feel that it has merit but does not fully meet PLOS ONE’s publication criteria as it currently stands. Therefore, we invite you to submit a revised version of the manuscript that addresses the points raised during the review process.

We look forward to receiving your revised manuscript.

Kind regards,

Hong Qin, PhD

Academic Editor

PLOS ONE

Journal Requirements:

4. Please update your submission to use the PLOS LaTeX template. The template and more information on our requirements for LaTeX submissions can be found at http://journals.plos.org/plosone/s/latex.

Additional Editor Comments (if provided):

Comments from the editorial office: We note that one or more reviewers has recommended that you cite specific previously published works. As always, we recommend that you please review and evaluate the requested works to determine whether they are relevant and should be cited. It is not a requirement to cite these works. We appreciate your attention to this request.

Reviewers' comments:

Reviewer's Responses to Questions

**Comments to the Author**

1. Is the manuscript technically sound, and do the data support the conclusions?

Reviewer #1: Yes

Reviewer #2: Yes

Reviewer #3: Yes

2. Has the statistical analysis been performed appropriately and rigorously? 

Reviewer #1: Yes

Reviewer #2: Yes

Reviewer #3: Yes

3. Have the authors made all data underlying the findings in their manuscript fully available?

Reviewer #1: Yes

Reviewer #2: Yes

Reviewer #3: Yes

4. Is the manuscript presented in an intelligible fashion and written in standard English?

Reviewer #1: Yes

Reviewer #2: Yes

Reviewer #3: Yes

5. Review Comments to the Author

Reviewer #1: Manuscript technically seems correct and the statistical analysis has been performed appropriately. It is presented in an intelligible fashion and written in standard English. Quantile regression model is useful technique. It is also available on research square.

Reviewer #2: This paper introduced a new bounded probability distribution derived from a transformation of the Weibull distribution. It examined key properties such as moments, entropy measures, and the quantile function. The authors developed the sequential probability ratio test (SPRT) and used the maximum likelihood estimation (MLE) method for parameter estimation. A Monte Carlo simulation assessed the performance of estimates. They also formulated a quantile regression model and applied it to real-world data on risk assessment and educational attainment. The results showed that the proposed model outperformed existing regression models in analyzing bounded variables.

However, the authors have considered a remarkable research work in a methodically structured manner. I have the following suggestions for improvement of the paper:

1. The authors should maintain uniformity in the use of abbreviations. If abbreviations are already defined, do not use the full form of them again.

2. Make the research gap section clear and highlight your work separately.

3. Please categorize your and previous research in the Table in the section Literature Review to show the better research gap.

Reviewer #3: Review Report of the Abstract

Title of the Abstract: New Bounded Unit Weibull Model: Applications with Quantile Regression

Authors: Laxmi Prasad Sapkota, Nirajan Bam and Vijay Kumar

Summary of the Abstract: This paper introduces a new bounded probability distribution derived from a transformation of the Weibull distribution. It discusses key properties such as moments, entropies, and the quantile function. Additionally, the sequential probability ratio test (SPRT) was developed for the proposed model. The maximum likelihood estimation method was used for parameter estimation, with Monte Carlo simulations assessing estimation performance. A quantile regression model was formulated and applied to data sets in risk assessment and educational attainment, showing superior performance over alternative regression models. The findings contribute to the statistical toolkit for analyzing bounded variables across scientific fields.

Strengths:

1. Relevance: The abstract addresses an important topic within its field.

2. Clarity: The research problem, methodology, and conclusions are clearly presented.

3. Novelty: The study provides new insights by introducing a new bounded probability distribution.

4. Methodological Rigor: The use of SPRT, maximum likelihood estimation, and Monte Carlo simulation strengthens the research.

Areas for Improvement:

1. Objective Statement: The purpose of the study could be more explicitly stated.

2. Comparison with Existing Models: While the study mentions alternative models, more details on their limitations and the advantages of the new model would enhance clarity. Authors may consider Toppe-Leone Distribution, Unit Gamma, Unit logistic etc.

Mazucheli, J., Menezes, A. F. B., & Dey, S. (2018). Improved maximum-likelihood estimators for the parameters of the unit-gamma distribution. Communications in Statistics-Theory and Methods, 47(15), 3767-3778.

Menezes, A. F. B., Mazucheli, J., & Dey, S. (2018). The unit-logistic distribution: Different methods of estimation. Pesquisa Operacional, 38(3), 555-578.

Shekhawat, K., Sharma, V.K. An Extension of J-Shaped Distribution with Application to Tissue Damage Proportions in Blood. Sankhya B 83, 548–574 (2021).

Mazucheli, J., Menezes, A. F. B., & Chakraborty, S. (2019). On the one parameter unit-Lindley distribution and its associated regression model for proportion data. Journal of Applied Statistics, 46(4), 700-714.

Recommendation:

• Accept.

6. PLOS authors have the option to publish the peer review history of their article (what does this mean?). If published, this will include your full peer review and any attached files.

Reviewer #1: No

Reviewer #2: No

Reviewer #3: No

---

## [Author Response · Author response to Decision Letter 1]

19 Mar 2025

We have uploaded the point-to-point responses to the reviewer's comments

---

## [Decision Letter · Decision Letter 1]

16 Apr 2025

New Bounded Unit Weibull Model: Applications with Quantile Regression

PONE-D-25-02426R1

Dear Dr. Sapkota,

We’re pleased to inform you that your manuscript has been judged scientifically suitable for publication and will be formally accepted for publication once it meets all outstanding technical requirements.

Kind regards,

Hong Qin, PhD

Academic Editor

PLOS ONE

Additional Editor Comments (optional):

Reviewers' comments:

Reviewer's Responses to Questions

**Comments to the Author**

1. If the authors have adequately addressed your comments raised in a previous round of review and you feel that this manuscript is now acceptable for publication, you may indicate that here to bypass the “Comments to the Author” section, enter your conflict of interest statement in the “Confidential to Editor” section, and submit your "Accept" recommendation.

Reviewer #1: All comments have been addressed

Reviewer #2: All comments have been addressed

Reviewer #3: All comments have been addressed

2. Is the manuscript technically sound, and do the data support the conclusions?

Reviewer #1: Yes

Reviewer #2: Yes

Reviewer #3: Yes

3. Has the statistical analysis been performed appropriately and rigorously? 

Reviewer #1: Yes

Reviewer #2: Yes

Reviewer #3: Yes

4. Have the authors made all data underlying the findings in their manuscript fully available?

Reviewer #1: Yes

Reviewer #2: Yes

Reviewer #3: Yes

5. Is the manuscript presented in an intelligible fashion and written in standard English?

Reviewer #1: Yes

Reviewer #2: Yes

Reviewer #3: Yes

6. Review Comments to the Author

Reviewer #1: Authors has revised the manuscript accordingly and uploaded the comments point to point. Now it is accepted for publication.

Reviewer #2: I dont have further comments.

The authors have incorporated all the comments/suggestions I raised in the previous version of the manuscript. I have no further comments. Therefore, this manuscript version may be accepted for publication in the esteemed journal, Plos One.

Reviewer #3: Authors have improved the article based on my previous suggestions. I have no further comments. Article may be accepted for its publ;ication.

7. PLOS authors have the option to publish the peer review history of their article (what does this mean?). If published, this will include your full peer review and any attached files.

Reviewer #1: No

Reviewer #2: No

Reviewer #3: No

---

## [Editor Report · Acceptance letter]

PONE-D-25-02426R1

PLOS ONE

Dear Dr. Sapkota,

I'm pleased to inform you that your manuscript has been deemed suitable for publication in PLOS ONE. Congratulations! Your manuscript is now being handed over to our production team.

Kind regards,

on behalf of

Dr. Hong Qin

Academic Editor

PLOS ONE